# The interplay of additivity, dominance, and epistasis on fitness in a diploid yeast cross

Takeshi Matsui[1,2,3,7], Martin N. Mullis [4,6,7], Kevin R. Roy [1,3,5], Joseph J. Hale[4], Rachel Schell[4], Sasha F. Levy [1,2,3 ✉] & Ian M. Ehrenreich [4 ✉]

In diploid species, genetic loci can show additive, dominance, and epistatic effects. To characterize the contributions of these different types of genetic effects to heritable traits, we use a double barcoding system to generate and phenotype a panel of ~200,000 diploid yeast strains that can be partitioned into hundreds of interrelated families. This experiment enables the detection of thousands of epistatic loci, many whose effects vary across families. Here, we show traits are largely specified by a small number of hub loci with major additive and dominance effects, and pervasive epistasis. Genetic background commonly influences both the additive and dominance effects of loci, with multiple modifiers typically involved. The most prominent dominance modifier in our data is the mating locus, which has no effect on its own. Our findings show that the interplay between additivity, dominance, and epistasis underlies a complex genotype-to-phenotype map in diploids.

---

[1] Joint Initiative for Metrology in Biology, Stanford, CA 94305, USA. [2] SLAC National Accelerator Laboratory, Menlo Park, CA 94025, USA. [3] Department of Genetics, Stanford University School of Medicine, Stanford, CA 94305, USA. [4] Molecular and Computational Biology Section, Department of Biological Sciences, University of Southern California, Los Angeles, CA 90089, USA. [5] Stanford Genome Technology Center, Stanford University, Palo Alto, CA 94304, USA. [6] Present address: Twist Bioscience, 681 Gateway Blvd, South San Francisco, CA 94080, USA. [7] These authors contributed equally: Takeshi Matsui, Martin N. Mullis. ✉email: sflevy@stanford.edu; ian.ehrenreich@usc.edu

Most complex traits, including many phenotypes of agricultural, clinical, and evolutionary significance, are specified by multiple loci[1]. How alleles at these loci collectively produce the heritable trait variation in genetically diverse populations remains unresolved[2,3]. While additive loci play a major role in most traits, non-additive genetic effects are also likely important[4–9]. However, loci with non-additive genetic effects are often difficult to detect, limiting knowledge of their properties[10,11].

The two purely genetic sources of non-additivity are dominance among alleles of individual loci and epistasis between alleles at different loci (or genetic interactions)[12,13]. Most empirical studies of non-additive genetic effects have focused on haploid or inbred individuals[14–16], which provide higher statistical power to detect loci due to their minimal levels of heterozygosity. However, by design, these populations cannot furnish insight into dominance or its relationship with epistasis. This is a problem because many eukaryotic species that matter to humans, including our species itself, exist predominantly as diploids that outbreed and have high levels of heterozygosity[17–19]. Dominance may be an important contributor to traits in these species.

When epistasis occurs in diploids, a locus may influence only the additive effects, only the dominance effects, or both the additive and dominance effects of its interactor(s)[20–22]. Such interplay has implications for efforts to genetically dissect phenotypes, predict heritable traits from genotypes, and understand the evolutionary trajectories of beneficial and deleterious alleles. Yet, exploration of the relationship between additivity, dominance, and epistasis has mainly been limited to theory because of technical challenges in identifying non-additive loci.

The budding yeast *Saccharomyces cerevisiae* is a potentially powerful system for studying non-additive genetics in diploids. Haploid yeast segregants with known genotypes can be mated to produce diploid strains that also have known genotypes[23]. This strategy facilitates the generation of diploid mapping populations that are roughly the square of the number of haploid progenitors. However, phenotyping large diploid populations of more than ~10,000 individuals has been technically difficult[23,24], limiting the use of this strategy.

Here, we develop a chromosomally-encoded barcoding system that enables phenotyping of large yeast diploid mapping populations. We fuse two genomic barcodes, one from each haploid parent, in vivo to create a unique double barcode for each diploid strain (Fig. 1a). This system enables linkage mapping in a population of ~200,000 diploid strains and examination of the relationship between additivity, dominance, and epistasis at detected loci.

## Results

### Phenotyping of a large diploid cross by barcode sequencing.
We started with two *S. cerevisiae* isolates, the commonly used lab strain BY4716 (BY) and a haploid derivative of the clinical isolate 322134S (3S) (Fig. 1a). These strains differ at ~45,000 SNPs (~0.4% of genome)[25,26]. To ensure segregation of the mating locus, both BY *MATa* x 3S *MATα* and 3S *MATa* x BY *MATα* crosses were performed using isogenic strains that had been mating type switched. From these crosses, 600 *MATα* and 400 *MATa* segregants from distinct four-spore tetrads were marked at the neutral *YBR209W* locus by integrating a random barcode[27,28]. At least two uniquely barcoded strains were recovered per haploid segregant and the genome of each segregant was sequenced to define the genotype represented by each barcode (Supplementary Figs. 1 and 2A).

*MATa* and *MATα* strains (2 barcodes per segregant) were mated as pairs and grown on media that induced site-directed recombination between the *MATa* barcode and *MATα* barcode on homologous chromosomes (Fig. 1b and Supplementary Fig. 1E)[28,29]. This process resulted in a double barcode on one chromosome that uniquely identifies both parents of a diploid strain and therefore its presumptive genotype (Supplementary Fig. 2B). Using similar methods, we also constructed BY/BY, BY/3S, 3S/BY, and 3S/3S parental diploid strains.

After the matings, diploid strains were pooled and competed in seven conditions: cobalt chloride, copper sulfate, glucose, hydrogen peroxide, sodium chloride, rapamycin, and zeocin with the glucose condition performed twice (Supplementary Table 1). Cells were grown for ~15 generations in serial batch culture, with 1:8 dilution every ~3 generations and a bottleneck population size greater than $2 \times 10^9$ cells (Fig. 1c). Double barcodes were enumerated over 4–5 timepoints by sequencing amplicons from the double barcode locus, and the resulting frequency trajectories were used to estimate the relative fitness of each strain[30,31].

We recovered on average 197,267 diploid strains per environment with a minimum of two biological replicates that were marked with different barcodes within a growth pool (Supplementary Table 2). These biological replicates showed a relatively high correlation (average Spearman's rho across environments = 0.67, 0.524 < rho < 0.8, Fig. 1d and Supplementary Fig. 3A). Noise between biological replicates was primarily caused by one or more replicates being present at a low abundance within the pool, resulting in a less accurate fitness estimate (Supplementary Fig. 4)[31]. To minimize the effects of measurement noise, the average fitness measure of all biological replicates was used as the phenotype for each strain. The average fitness measures of strains assayed in replicate glucose growth cultures were also highly correlated (Spearman's correlation = 0.863, Fig. 1e). This suggests that our pooled fitness assay is accurate and reproducible.

Substantial phenotypic diversity was observed in every environment. The majority of this variation was due to genetic factors: broad-sense heritabilities were on average 61% (52–76% across environments), with 40% (19–53%) being additive and 21% (19–26%) being non-additive (Fig. 1f, Supplementary Fig. 5, and Supplementary Table 3). These heritability estimates are similar to other yeast studies in which fitness was measured using colony growth assays on agar plates[2,8,26]. Every environment contained many diploids with more extreme fitness than either the BY/BY or 3S/3S parent (i.e., transgressive segregation). BY/3S and 3S/BY segregants were more fit than the BY/BY or 3S/3S diploids in all environments but one (i.e., heterozygote advantage, Fig. 1g).

### Genetic mapping within interrelated families.
Using quantile normalized fitness estimates from barcode sequencing, we mapped loci that contribute to growth (Supplementary Fig. 6). Due to our experimental design, diploid strains generated from the same haploid parent (families) are more genetically related than diploid strains generated from different parents (Fig. 1a). Such family structure causes false positives in genetic mapping[32,33]. Here, we found that most sites throughout the genome exceeded nominal significance thresholds when fixed effects linear models were applied in a given environment (Supplementary Fig. 7). To enable mapping despite the family structure, we used mixed effects linear models, which are commonly employed in genetic mapping studies involving populations in which individuals show non-random relatedness[34–36]. Specifically, we used Factored Spectrally Transformed Linear Mixed Models (FaST-LMM)[37,38] to identify an average of 18 loci per environment (10–26).

Contrary to expectations that larger sample sizes should yield better statistical power and therefore more detections, the numbers of loci identified here were comparable to studies that

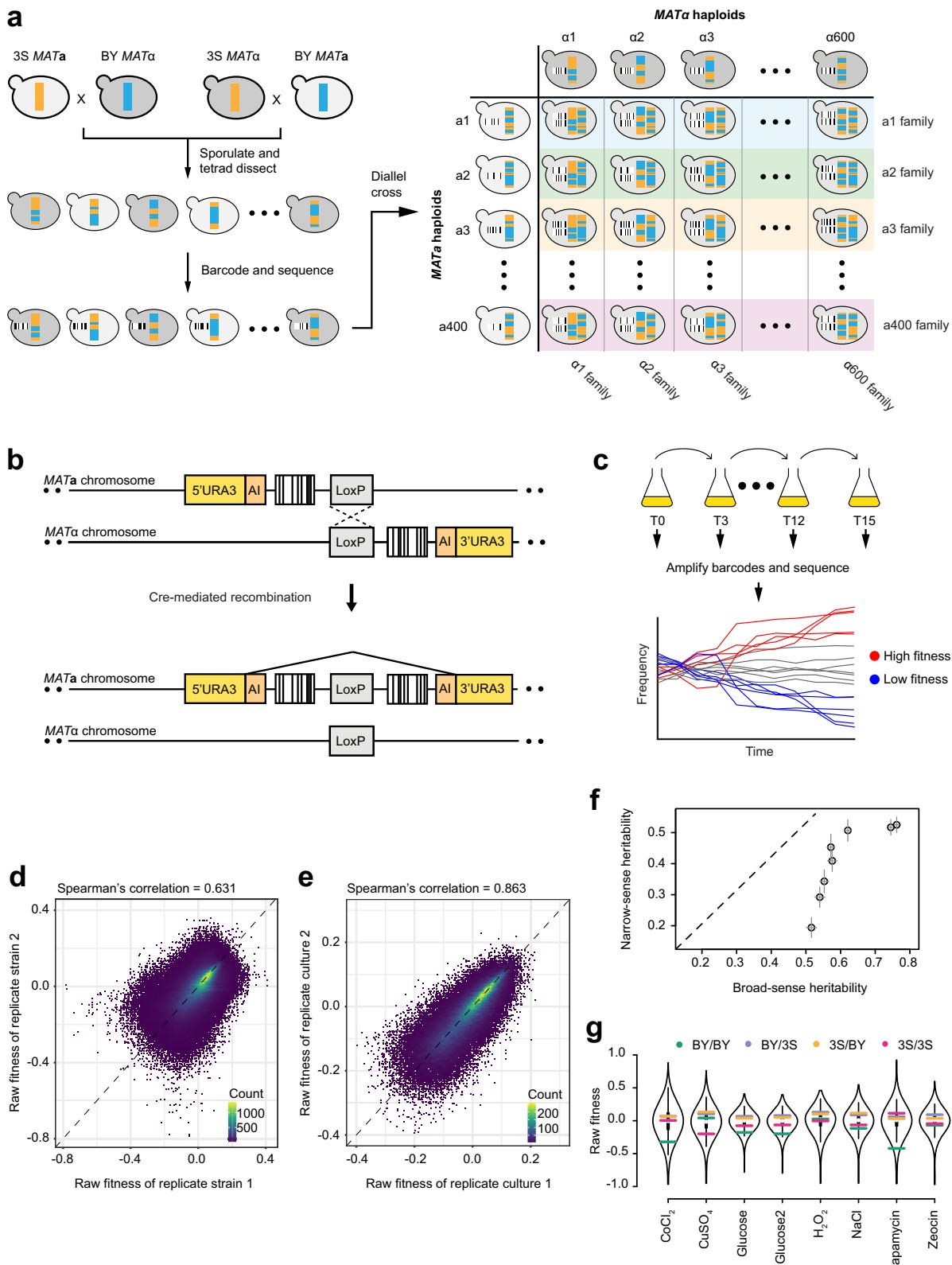

were at least 60-fold smaller[2,14,23]. To identify more loci, we also used an alternative strategy that did not require explicitly controlling for family structure. Fixed effects linear models were conducted individually within each of 392 families of diploids that descended from distinct *MAT***a** parents and consisted of ~600 individuals each. The family-level scans yielded an average of 6.5 detections per family across environments (Fig. 2a, b and

Supplementary Fig. 8), which were largely reproducible between replicate glucose cultures (Fig. 2c).

Often the same loci were detected in multiple families within an environment, as expected if many detections within families are true positives. Detections across the families were consolidated, resulting in approximately 58 distinct loci per environment (49–65), >2.5-fold more loci than detected by FaST-LMM (Fig. 2a and b). Across

**Fig. 1 Generating a large panel of diploid segregants with known genotypes that can be phenotyped as a pool. a** Overview of the experimental design. Parental haploids, BY and 3S, were mated and sporulated. The resulting $MAT\alpha$ and $MATa$ segregants were barcoded at a common genomic location and sequenced. Segregants were mated as pairs to generate a panel of ~200,000 double-barcoded diploid strains with known genotypes. All diploid strains originating from a single haploid parent are referred to as a 'family'. **b** $MAT\alpha$ and $MATa$ barcodes were brought to the same genomic location by inducing recombination between homologous chromosomes via Cre-loxP. **c** Diploid strains were pooled and grown in competition for 12–15 generations. Barcode sequencing over the course of the competition was used to estimate the fitness of each strain. **d** Density plot of the raw fitness of double barcodes representing the same diploid strain in the same pooled growth condition (Glucose 1). **e** Density plot of the mean raw fitness of the same diploid strain measured in two replicate growth cultures (Glucose 1 and Glucose 2). **f** The mean broad-sense and narrow-sense heritability estimates for the 8 environments. The standard errors for both heritability estimates are shown as error bars for each point. **g** Violin plots of the fitnesses of diploid strains in 8 environments ($n > 187,000$ in each environment). Raw fitness estimates of BY/BY, BY/3S, 3S/BY, and 3S/3S diploid strains are shown as colored lines. Overlaid boxplots, here and in subsequent figures, indicate the median (white dots), interquartile range (IQR; black boxes), and lower and upper adjacent values (black lines extending from the black boxes), defined as first quartile $-$ 1.5 IQR and third quartile $+$ 1.5 IQR, respectively.

environments, these distinct loci were detected in as few as one family and as many as 388 families (Fig. 2d). To distinguish loci most likely to represent true positives, we identified distinct loci showing more detections across families than expected by chance using an approach first employed to map loci with widespread effects on transcription[39]. We found on average 22 of these enriched loci per environment (15–30; Fig. 2e).

Many of the loci identified by FaST-LMM overlapped with the conservative set of enriched family-level loci, but there were also a number of differences between the methods (Fig. 2d). 29% of loci detected by FaST-LMM were detected in fewer families than the enrichment threshold in a given environment (10% to 38%; Fig. 2e; Supplementary Fig. 9). Similarly, 43% of loci enriched across families were not identified by FaST-LMM (35% to 57%; Fig. 2e). These findings likely reflect numerous complexities in our diploid mapping population, including varying degrees of relatedness among strains within and between families and a potentially large number of segregating loci that vary in their effect sizes, dominance, epistasis, and linkage to each other. Neither FaST-LMM nor family-level scans can fully address all these factors and thus they may produce somewhat different results, suggesting the two methods should be viewed as complementary.

**Loci frequently show dominance effects**. In diploids, non-additivity can arise due to dominance among alleles at the same locus, epistasis between alleles at different loci, or a mixture of the two. To identify such non-additive loci from the aggregate data, we extracted the non-additive portion of each diploid strain's phenotype (Supplementary Fig. 10)[23]. Using these values accounts for family structure and enables mapping of non-additive loci in the full segregant panel with fixed effects linear models. Regarding dominance effects, we identified an average of 18 loci showing dominance per environment (12–30). 49% of these loci were also identified by FaST-LMM, while 72% of these loci were detected in the conservative set of loci from the family-level scans. Among loci with dominance effects, the average degree of dominance was ~51% (i.e., heterozygotes' fitnesses were roughly halfway between the average of the two homozygotes and one of the homozygotes), with 82% of the loci showing incomplete dominance (Fig. 2f, g). Only ~7% of the loci exhibited complete dominance, while overdominance (~8%) and under-dominance (~3%) were seen among the remaining loci with dominance effects. ~77% of the loci showed dominance towards the allele conferring higher fitness (Fig. 2g), which may explain why segregants were more fit than the BY/BY or 3S/3S diploid strains (Fig. 1g).

**Epistatic hubs govern both additivity and non-additivity**. We also used the non-additive portion of phenotype to perform

comprehensive genome-wide scans for genetic interactions. We identified an average of 440 two-locus interactions per environment (377–538; Fig. 3a and Supplementary Fig. 11). Our large sample size had a pronounced impact on detection: ~40-fold more interactions per environment were detected than previous studies that phenotyped smaller mapping populations using conventional approaches[14,23]. Our large sample size also enabled comprehensive scans for three-locus interactions with a reduced set of markers, identifying an average of 6152 per environment (4845–7301; Fig. 3a and Supplementary Fig. 12). Loci involved in three-locus interactions were identified across all chromosomes and distributed widely throughout the genome.

We next analyzed the relationship between individual loci and their genetic interactions. We found a strong positive relationship between the effect of a locus and its involvement in two- and three-locus interactions (Fig. 3b). This suggests that loci with larger effects tend to genetically interact with many loci or that their interactions are easier to detect. We also observed a clear linear relationship between the number of two- and three-locus interactions of a given locus (Fig. 3c). Notably, certain loci exhibited many more interactions than others, acting as 'hubs' (here defined as loci with >20 two-locus interactions in at least one environment)[8]. On average, ~4.5 hubs were detected per environment, and the same hub was often detected in multiple environments. A majority (>54%) of all two- and three-locus interactions involved at least one hub. Fine-mapping localized the Chromosome VI, VIII, X, and XII hubs to genes involved in amino acid sensing (*PTR3*), copper resistance (*CUP1*), vacuolar protein sorting (*VPS70*), and a gene of unknown function (*YLR257W*), respectively.

**Relationships between epistasis and dominance in diploids**. In haploids, epistasis can only influence the additive effect of a locus because there are no heterozygotes. In diploids, however, epistasis can modify a locus' additive effects, dominance effects, or both additive and dominance effects[20–22]. To better characterize how loci are modified, each two-locus interaction was partitioned into additive and dominance components. We found that changes in dominance account for ~44% of the average epistatic effect (Fig. 3d, e), implying that interactions often affect both additivity and dominance. However, this fraction varied depending on whether the modifying locus was a hub. When the modifier was a hub, dominance accounted for little of the epistatic effect (11.9% on average), implying that hubs mostly modify the additive component of the interacting loci. By comparison, when the modifier was not a hub, epistasis was mostly composed of dominance (64% of interactions had a larger dominance component). These data suggest that epistasis commonly involves modification of dominance in diploids and that hubs act in a distinct manner from loci that are not hubs.

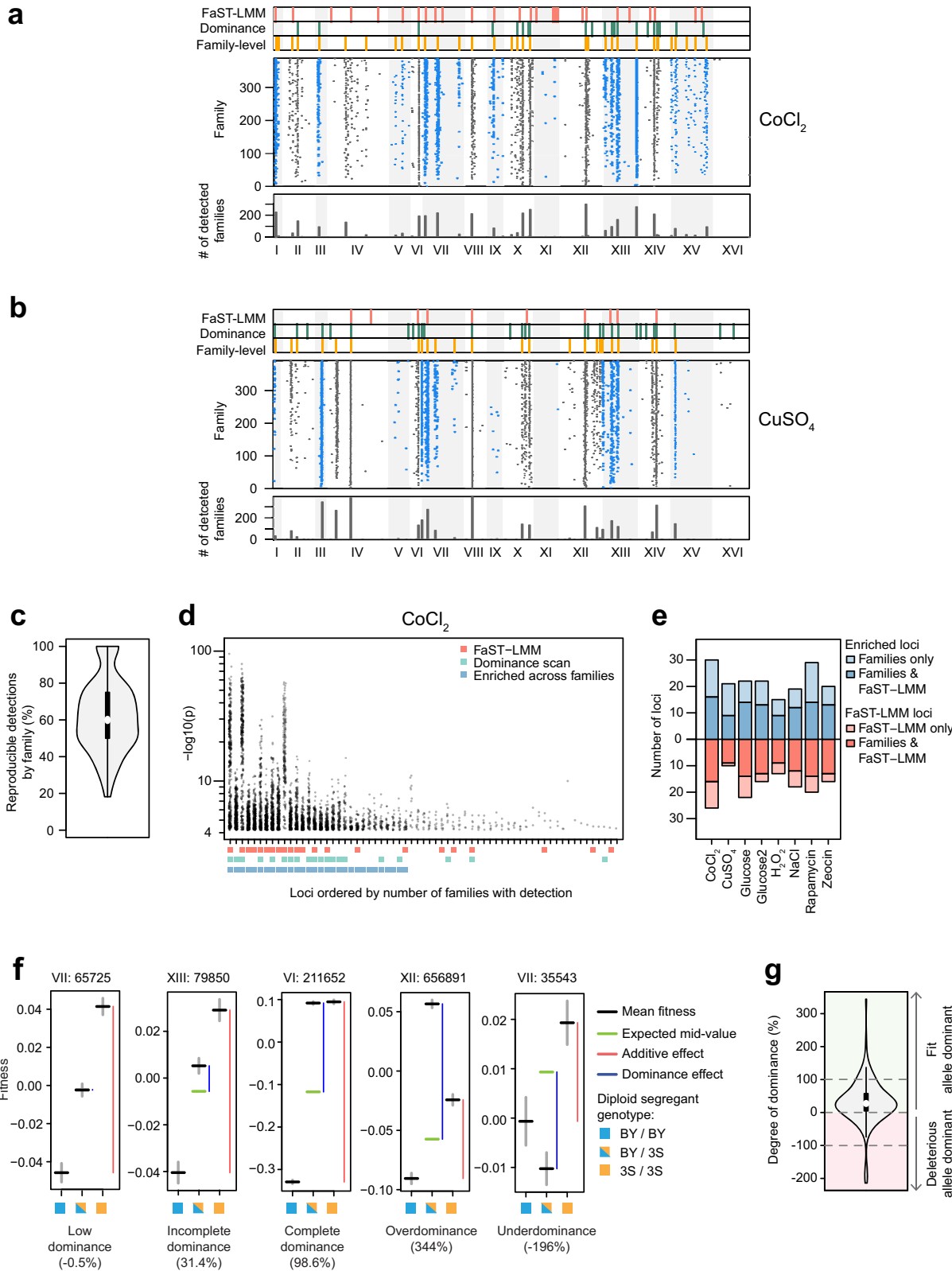

We next examined how the additive and dominance effects of hubs were modified by genetic interactions. In most cases, hubs genetically interacted with a small number of major effect modifiers and many minor effect modifiers (Fig. 4a). The major effect modifiers typically influenced only the additive or only the dominance effect of a hub, suggesting that distinct sets of loci govern additive and dominance effect sizes

(Fig. 4a). Whereas the most frequent major effect modifiers of the additive effects of hubs were other hubs (Fig. 4b), the single most frequent major effect modifier of the dominance effects of hubs was a locus on Chromosome III. Collectively, multiple modifier loci could cause a hub locus to show a broad range of effect sizes across different genetic backgrounds (Fig. 4c).

**Fig. 2 Identification of loci that affect fitness. a, b** Loci mapped in $CoCl_2$ (**a**) and $CuSO_4$ (**b**). Panels from top to bottom are (1) loci detected using the mixed effects linear model FaST-LMM (red bars), (2) loci with dominance effects detected using a fixed effects linear model on the non-additive portion of each diploid's phenotype (green bars), (3) loci enriched for detections in family-level scans (orange bars), (4) loci detected using family-tests (black or blue points), where each row is a different *MATa* family, and (5) the total number of detections across families for each 20 kb interval (gray bars). **c** Violin plot showing the % of loci that were detected in both glucose replicates for each family ($n = 392$ *MATa* families). **d** Scatterplot showing distinct loci detected using family-level tests with permutation-based thresholds in $CoCl_2$ and their maximal $-\log_{10}(p)$ values in each family in which they were identified. Red, green, and blue labels denote distinct loci in family-level scans that were also identified by FaST-LMM, dominance scans, or enrichment tests, respectively. Distinct loci showed substantial variability in statistical significance across families and mapping methods. **e** Barplot comparing the number of enriched family-level loci (red) and FaST-LMM loci (blue) detected across environments. Loci that were detected using both mapping methods are in dark colors, while loci that were specific to either FaST-LMM or family-level scans are in light colors. **f** Examples of loci with only additive effects (or low dominance), incomplete dominance, complete dominance, overdominance, and underdominance. All genotype classes had $n > 41,000$. Black lines are the mean fitness of diploids subsetted by the genotype state at the focal locus. Gray lines are the standard errors. Green lines are the expected mean fitness of heterozygotes assuming no dominance. Genotype state at each locus is denoted by colored boxes: BY/BY (blue), 3S/3S (orange), is BY/3S (half blue, half orange). Dominance and additive effects (blue and red bars, respectively) for each subset of the data are shown next to the relevant genotype classes. The degree of dominance at a locus is included in parentheses. **g** Violin plot showing the degree of dominance for all loci detected in the dominance scan ($n = 142$). Loci with positive values are dominant towards the allele conferring higher fitness (green), while loci with negative values are dominant towards the deleterious allele (red). All loci with degree of dominance >100% or < −100% exhibit overdominance and underdominance, respectively.

---

**Characteristics of the Chromosome III dominance modifier.** Although not a hub, the Chromosome III locus nevertheless had a prominent impact on phenotype by modifying the dominance effects of multiple variable effect loci. Interactions with the Chromosome III locus had greater impacts on dominance than additivity at all focal loci. For example, in hydrogen peroxide, dominance at the Chromosome X variable effect locus depended on the Chromosome III locus, ranging from complete to nearly absent in a genotype-dependent manner (Fig. 5a). We delimited the Chromosome III locus to a 3 kb region containing the mating locus and a few other genes (*BUD5*, *TAF2*, and *YCR041W*).

Yeast mating types possess different nonhomologous gene cassettes at the mating locus, which encode distinct transcription factors that are master regulators of the *MATa*, *MATα*, and diploid transcriptional programs[40]. This region of the genome is unique because four genotype classes segregate (BY *MATa*, 3S *MATa*, 3S *MATα*, and BY *MATα*), and as a result, the two heterozygotes are not identical (Supplementary Fig. 13). To test if the mating locus is the dominance modifier, we partitioned heterozygotes at the Chromosome III locus based on their parents-of-origin for the *MATa* and *MATα* cassettes and found a difference: dominance at the Chromosome X locus was only visible in the 3S *MATa*/BY *MATα* genotype class (Fig. 5a). Other hub loci modified by Chromosome III showed the same relationship between dominance and the parent-of-origin of the mating loci (Fig. 5b). These results suggest that BY and 3S harbor functional differences in one or both mating cassettes.

## Discussion

We used a double barcoding system to generate and phenotype an extremely large panel of diploid yeast strains that can be partitioned into hundreds of interrelated families. This experimental design enabled the detection of thousands of loci, including at least an order of magnitude more genetic interactions than discovered in previous yeast crosses. Analysis of these epistatic loci identified a modest number of hubs that have large effects, show pervasive epistasis, and control most phenotypic variation across environments, as well as many other loci that genetically interact with these hubs.

Genetic background commonly modified the magnitude of, or completely masked, the effects of the hubs, indicating that even loci with the largest effects are highly sensitive to genetic background. Such non-additive genetic background effects are likely to hamper efforts to predict phenotype from genotype by limiting the extrapolation of effect estimates from one genetic context to others. However, our finding that large effect loci were most impacted by other major effect loci does provide some optimism that characterizing a limited set of interactions may account for a substantial portion of these genetic background effects.

Because our experiments were performed in outbred diploids rather than haploids or inbred diploids, we could detect dominance effects and whether dominance is modified by epistasis. We showed that dominance effects are common and that the magnitude of dominance can strongly depend on the alleles of interacting loci. The potential existence of dominance modifiers has been discussed in theory, but to date, only a single dominance modifier has been found in a plant self-incompatibility locus[41,42]. Our results show that dominance modifiers are prevalent and raise the intriguing possibility that sites with atypical allele dynamics within natural populations, the yeast mating locus here and a self-incompatibility locus in plants, are more likely to harbor dominance modifiers with major effects.

Generally, we found that heritable traits in yeast are more genetically complex than formerly appreciated. Relative to the cross that we examined, natural populations may harbor substantially higher genetic diversity, meaning traits could be even more complex and difficult to dissect. Our work supports the premise that, to the extent possible, focusing on groups of more closely related individuals, such as the families studied here, can enhance statistical power and precision relative to populations with greater diversity[23,24,43]. The genetic insights gained from these more closely related groups can then be leveraged to inform the genetic architecture of traits in more diverse populations in which many critical genetic effects may otherwise be obscured.

## Methods

**Generation of haploid segregants.** All haploid segregants and diploid segregants described in this paper were generated from a cross using two isolates of *Saccharomyces cerevisiae*, the commonly used lab strain BY4716 (BY) and a haploid derivative of the clinical isolate 322134S (3S). To generate counterselectable markers in each strain, we first introduced clean deletions of *FCY1* and *URA3*. Each gene was deleted using a two step approach: first, genes were replaced with a *KanMX* cassette[44] via lithium acetate transformation[45]. Next, the cassette was targeted using CRISPR/Cas9 and a gRNA specific to the *pTEF* promoter region in each cassette[46]. A repair template homologous to the upstream and downstream region of the target gene was co-transformed with the CRISPR system to generate a clean deletion. The *MATa* BY and 3S strains were then mating-type-switched by transforming a plasmid containing galactose-inducible *HO* and *URA3* using the lithium acetate protocol[47]. Strains with the plasmid were selected on SCM-Ura plates and inoculated into SCM-Ura + 2% galactose media overnight. Individual colonies were then obtained by plating out $10^2$ cells onto YPD plates. Colonies were tested for their mating type using the yeast mating halo assay[48]. Successfully mating-type-switched BY and 3S *MATα* clones were cured of the *HO* plasmid by

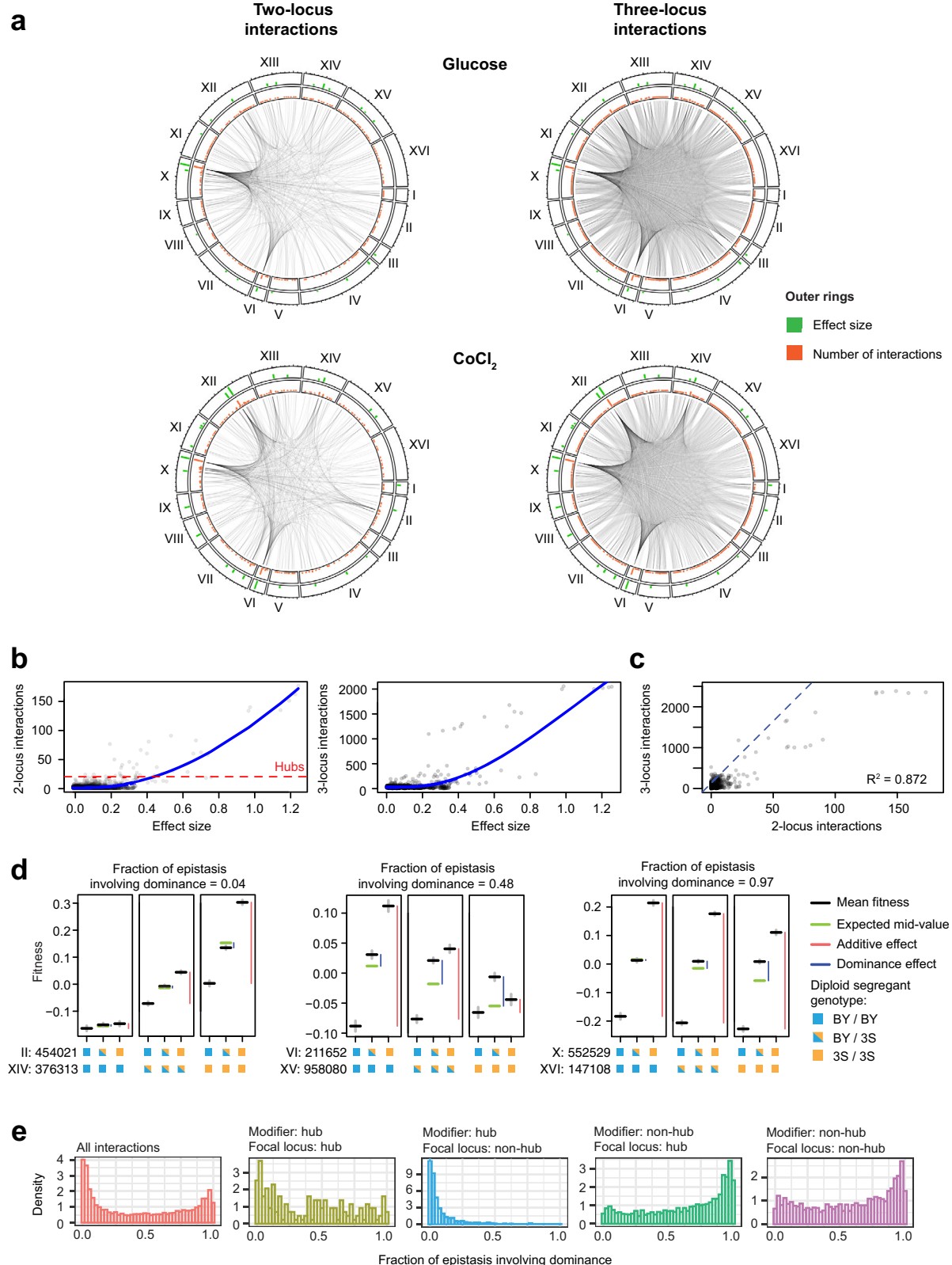

growing the cells on YPD + 5-FOA plates. The BY *MATa* and *MATα* cross parents were preserved at −80 °C with the unique identifiers IEY1176 and IEY1177, respectively. The 3S *MATa* and *MATα* cross parents were preserved at −80 °C with the unique identifiers IEY1178 and IEY1179, respectively.

To prevent aggregation of cells in pools grown in liquid, *FLO8*, a transcriptional activator of many flocculins[49], and *FLO11*, the flocculin responsible for many aggregation phenotypes in *S. cerevisiae*[50], were first knocked out in both mating

types of the BY *fcy1Δ ura3Δ* and 3 S *fcy1Δ ura3Δ* strains. Each parental strain was then engineered to have a genomic 'landing pad'[27–29] containing two partially crippled LoxP sites[51], Lox71 and Lox2272/71, and a galactose-inducible Cre recombinase[52] at the *YBR209W* locus via CRISPR/Cas9-mediated homologous recombination (Supplementary Fig. 1A). Earlier studies have shown that deletion of *YBR209W* or incorporation of our barcoding system has no effect on fitness[27].

**Fig. 3 Interactions often affect both the additive and dominance effects of involved loci. a** Interaction plots of all two-locus (left) and three locus (right) effects for two representative environments. Significant interactions between loci are shown as connecting lines. Green bars are the absolute effect size of a locus, calculated as the absolute difference between the mean fitness of diploids that are 3S/3S and BY/BY at the focal locus. Orange bars are the number of interactions detected for each locus. **b** Scatter plot of the absolute effect size of a locus and the number of two-locus (left) and three-locus (right) interactions in which it is involved. Local regressions are shown as blue lines. **c** Scatter plot of the number of two-locus and three-locus interactions per locus. **d** Examples of genetic interactions with, from left to right, low (0.04), moderate (0.48), and high (0.97) fractions of epistasis involving dominance. All genotype classes had $n > 8,700$. Black lines are the mean fitness of diploids subsetted by the genotype state at the two involved loci. Gray lines are the standard errors. Green lines are the expected mean fitness of heterozygotes assuming no dominance. Genotype state at each locus is denoted by colored boxes: BY/BY (blue), 3S/3S (orange), is BY/3S (half blue, half orange). The first locus is the locus whose effect is being modified, and the second locus is the modifier locus. Dominance and additive effects (blue and red bars, respectively) for each subset of the data are shown next to the relevant genotype classes. **e** Density plot of the fraction of epistasis involving dominance for all interactions (red; $n = 3,522$ genetic interactions), hub-hub (yellow; $n = 87$), non-hub-hub (blue; $n = 2197$), hub-non-hub (green; $n = 2197$), and non-hub-non-hub interactions (purple; $n = 1,238$).

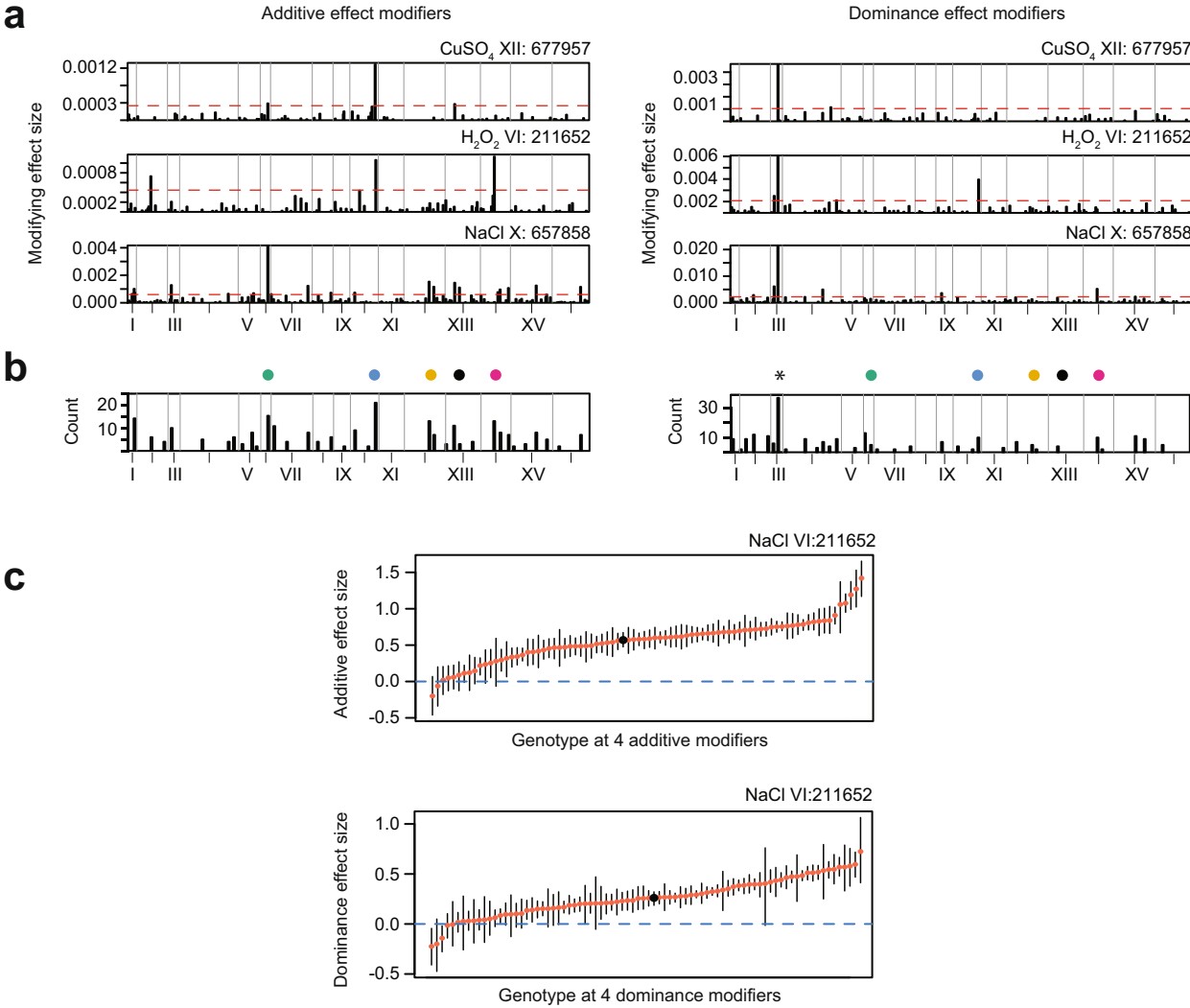

**Fig. 4 Multiple modifier loci cause hubs to exhibit a range of effect sizes across different genetic backgrounds. a** Specific examples of hubs on Chromosome VI, X, and XII (each row) and their additive (left) and dominance (right) modifiers. The height of the bar corresponds to the magnitude of the modifying effect. The dotted red line shows the threshold in which loci were considered as major effect modifiers. **b** Barplot showing the total number of times loci were detected as a major effect modifier of additive (left) or dominance (right) effects of hubs across environments. Colored dots indicate hub loci. An asterisk indicates a non-hub locus on ChrIII. **c** Additive (top; $n > 443$ for each genotype class) or dominance (bottom; $n > 175$ for each genotype class) effect size of Chromosome VI hub in NaCl across different allelic combinations of its four largest effect modifiers. Red points are the mean effect size of a genotype class based on the genotype state of the four modifiers. Black point is the overall mean effect size of the locus. Black lines are bootstrapped 95% confidence intervals.

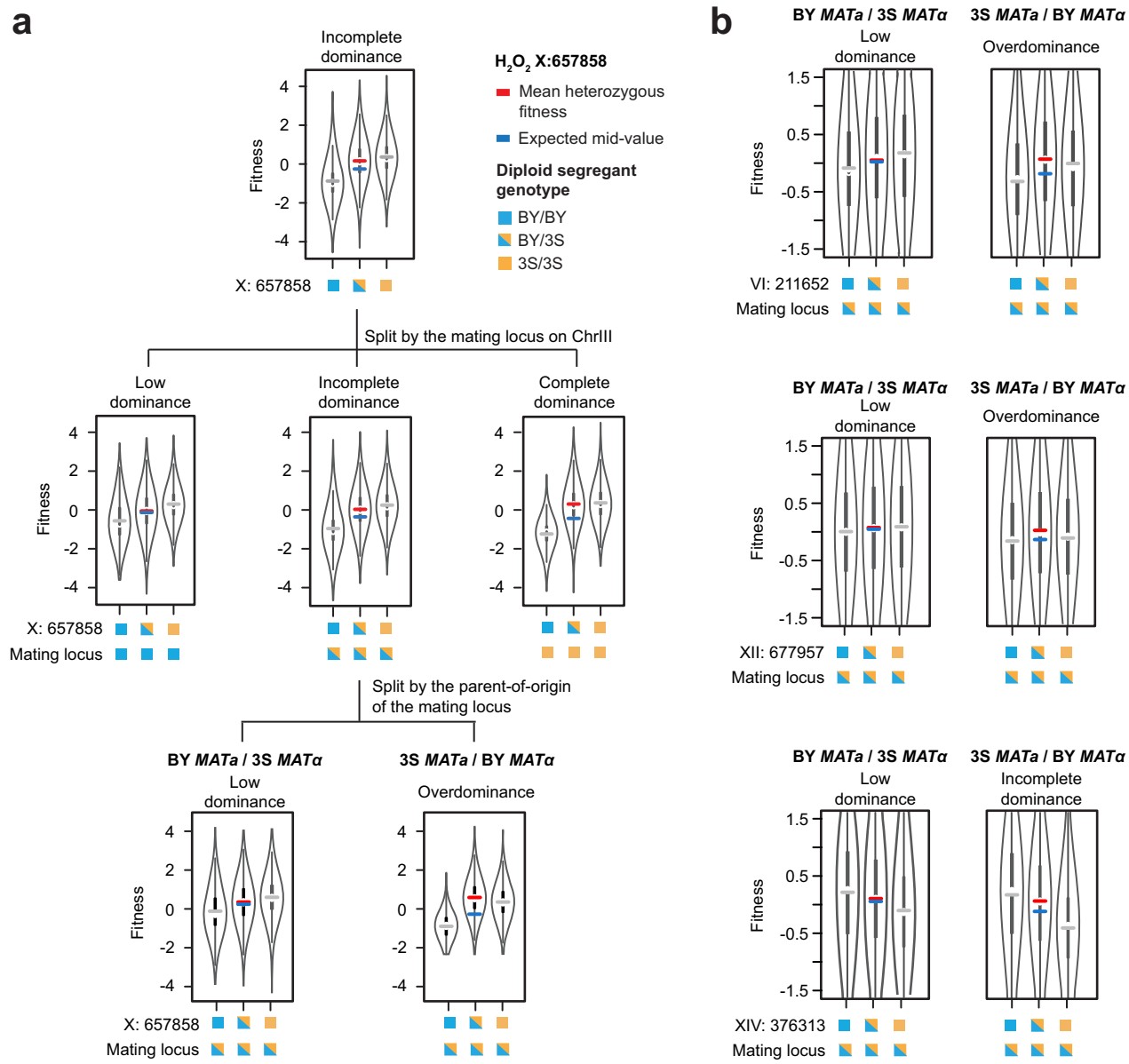

**Fig. 5 Parent-of-origin of the mating locus influences dominance at hubs. a** Violin plots of the fitness distribution of diploid strains split by the genotype at the Chromosome X locus (top), further split by the genotype at the mating locus on Chromosome III (middle), and the parent-of-origin at the mating locus (bottom). All genotype classes had n > 9000. Genotype state at each locus is denoted by colored boxes: BY/BY (blue), 3S/3S (orange), is BY/3S (half blue, half orange). Lines are the observed mean fitness in the homozygous genotype classes (gray), the observed mean fitness in heterozygous genotype classes (red), and the expected heterozygous fitness if there was no dominance (blue). **b** Violin plots of the fitness distribution of diploid strains split by the genotype state of a hub locus and the parent-of-origin of the mating locus: Chromosome VI hub (top), Chromosome XII hub (middle), Chromosome XIV hub (bottom). All genotype classes had n > 9500.

Opposite mating types of BY *fcy1Δ flo8Δ flo11Δ ura3Δ ho YBR209W::pGal1-Cre - Lox71 - Lox2272/71* and 3S *fcy1Δ flo8Δ flo11Δ ura3Δ ho YBR209W::pGal1-Cre - Lox71 - Lox2272/71* were next mated, creating two BY/3S heterozygous diploids. Diploids were sporulated and >500 tetrads were dissected from each diploid. Performing these two crosses would, in theory, enable us to achieve ~50% allele frequency at all sites, including the mating type locus (Supplementary Fig. 2). Either one *MATa* segregant or one *MATα* segregant was then randomly selected from each tetrad (a total of >500 each) to maximize the number of unique recombination breakpoints. To avoid segregants carrying aneuploidies, only tetrads that produced 4 spores were utilized.

**Barcoding of haploid segregants**. Segregants were uniquely barcoded using two different methods (Supplementary Fig. 1A–D). *MATα* segregants were barcoded by integrating a randomly barcoded plasmid via Cre-mediated homologous recombination at lox2272/71 (Supplementary Fig. 1C). Barcoded plasmids were made by modifying the pBAR6 plasmid[28]. First, pBAR6 was digested with KpnI and EcoRI

(Supplementary Fig. 1B). Linearized pBAR6 was then assembled by Gibson assembly with a PCR product containing a Lox2272/66 site, a random 20-mer barcode sequence, and a partial TruSeq read 2 adapter sequence. The resulting product was transformed into chemically competent NEB 10-beta cells using standard heat shock protocol, and transformants were selected on LB + 50 μg/ml Carbenicillin plates. To ensure high barcode complexity, ~100,000 transformants were scraped and pooled prior to plasmid extraction. Purified barcoded plasmids (300 ng) were then transformed into the yeast cells using the lithium acetate protocol[45]. After transformation, the barcoded plasmids were recombined into the yeast genome by inducing the galactose-inducible Cre recombinase by growing the yeast cells in YP + 2% galactose media. Homologous recombination between the two partially crippled Lox2272 variants, Lox2272/66 and Lox2272/71, resulted in the formation of a fully crippled Lox2272/66/71 and a fully functional Lox2272. Transformants with successful integration were selected on YPD + 200 μg/ml G418 agar plates.

*MATa* segregants were barcoded by CRISPR/Cpf1-driven[53] homologous recombination at the genomic landing pad (Supplementary Fig. 1D). The genomic

region containing the two partially crippled Lox sites, Lox71 and Lox2272/71, was replaced with a DNA fragment containing (in the following order) a 60 bp sequence homologous to the region upstream of the Lox sites, an HphMX cassette[54], the 5′ end of a split URA3 marker, a 5′ artificial intron splice site, a partial TruSeq read 1 adapter sequence, a random 20-mer barcode sequence, a partially crippled Lox66 site, and a 60 bp sequence homologous to the region downstream of the Lox sites. To integrate the randomly barcoded DNA fragment in the yeast genome, 200 ng of the DNA fragment, 200 ng of a PCR amplicon containing pTEF-CPF1 flanked by 2 nuclear localization sequences, and 200 ng of a PCR amplicon containing a polyA-tailed pSNR52−20-mer guide sequence were co-transformed into the yeast cells using lithium acetate[45]. Transformants with successful integration were then selected on YPD + 300 μg/ml Hygromycin B agar plates.

For each segregant transformation, we chose ~5 colonies that presumably represented independent integration events and barcodes. In addition to the segregants, both mating types of the parental strains were barcoded in the same manner as the segregants and ~50 colonies were picked for each parental strain.

**Whole-genome sequencing of parental haploid segregants**. Clones containing different barcodes for 1003 MATα and 500 MATa segregants were pooled and these pools were whole-genome sequenced at low-coverage (~10×) to determine where crossover events occured. A sequencing library was prepared using the Illumina Nextera Kit with custom multiplexing barcodes[2,15,26]. Libraries from different segregants were pooled in equimolar fractions and these multiplex pools were size selected using the Qiagen Gel Extraction Kit. Multiplexed samples were then sequenced on an Illumina HiSeq 2500 using 150 bp × 150 bp paired-end reads. For each strain, reads were mapped against the S288c genome using BWA with default settings[55]. Alignments were converted to a bam format and sorted using SAMTOOLS (default settings)[56]. Read duplicates were then removed and bam files were converted to pileups using SAMTOOLS (default settings). Base calls and coverage values were obtained from the pileup files for 43,865 high-confidence SNPs that segregate in the BYx3S cross. Any segregants that showed signs of cross-contamination, where the allele frequency at the SNP markers significantly deviated from the expected 0% or 100% 3S allele, were excluded from further analysis. To avoid segregants with aneuploidy, all segregants whose average coverage of each individual chromosome or segments of the chromosome significantly deviated from the overall average coverage were also removed. Additionally, all segregants with a mean coverage of less than 2 were removed. In total, 76 MATα and 11 MATa segregants were removed after filtering for quality. For the remaining 927 MATα and 489 MATa segregants, a vector containing the fraction of 3S calls at each SNP was generated and used to make initial genotype calls with sites above and below 50% classified as 3S and BY, respectively. This vector of initial genotype calls was then corrected with a Hidden Markov Model (HMM), implemented using the HMM package version 1.0[57] in R[58]. We used the following transition and emission probability matrices: transProbs = matrix(c(0.9999, 0.0001, 0.0001, 0.9999), emissionProbs = matrix(c(.0.25, 0.75, 0.75, 0.25). All SNP markers within the first and last 30 kb of each chromosome were omitted because we observed higher sequencing error rate and/or lower read mapping quality for these specific regions. Adjacent SNPs in the HMM-corrected genotype calls that lack recombination in the segregants were collapsed into a single SNP, reducing the number of SNP markers in subsequent analysis from 43,865 to 7742.

**Determining the barcode sequences**. For all segregants, the associated barcodes were determined by pooling all clones and sequencing the genomic region containing the barcode using Novogene Illumina HiSeq 2500 150 bp × 150 bp paired-end reads at ~2000× coverage per barcode. For the MATa segregants, the genomic region containing the barcode was PCR amplified using primer pairs where one primer was located on the partial TruSeq read 1 adapter sequence and the other primer was located downstream of the barcode. Similarly, barcodes for MATα segregants were PCR amplified using primer pairs where one primer was located on the partial TruSeq read 2 adapter sequence and the other primer was located upstream of the barcode. PCR products were purified using a Qiagen MinElute PCR purification kit, amplified using Illumina P1 and P2 primers, and then size selected via gel extraction prior to sequencing. The 20-mer barcode sequences for each segregant were then extracted from the sequencing reads and barcodes within a Hamming distance of 2 were clustered with Bartender[30]. Only barcode clusters comprising >5% of the total reads for each sample were considered true barcodes. One possible concern with using random barcodes is that sequencing errors from an abundant barcode could erroneously contribute to read counts of a barcode with a similar sequence. To prevent this, we determined the number of mismatches away from a nearest neighbor for each barcode. All barcodes were at least 4 mismatches away from each other. Thus, a sequencing read was unlikely to be assigned to the wrong barcode cluster unless it contained 3 or more errors. Overall, we recovered 403 MATa segregants and 679 MATα segregants with good genotype calls and at least 3 barcodes that are unique from all others.

**Generation of diploid segregants**. 400 MATa and 600 MATα segregants were mated to create a panel of ~240,000 diploid segregants, each labeled with 4 unique pairs of barcodes (~960,000 total double barcodes). To minimize skews in the

initial frequency distribution of genotypes, each MATa segregant was individually mated to each MATα segregant to generate the panel of diploid segregants. Specifically, two barcoded versions of each segregant were first grown to saturation and mixed in equal proportion. Each MATa segregant mixture was then systematically mated with a MATα segregant mixture using a LabCyte Echo 550 acoustic liquid handling robot. For each pairwise mating, the Echo transferred 100 nl of MATa and MATα segregants onto overlapping positions on a new YPD plate, resulting in the formation of diploid segregants labeled with ~4 double barcodes. After growing the cells overnight at 30°C, successfully mated diploids were isolated from unmated haploids by pinning the colonies onto YPD + 200 μg/ml G418 + 300 μg/ml Hygromycin B agar plates using the Singer ROTOR pinning robot and grown for 2 days. Haploid strains carried either KanMX or HphMX, therefore only mated diploids that carry both selection markers should grow. In addition to the diploid segregants, different barcoded versions of the homozygous and heterozygous parental diploid strains were made as controls, using the same approaches described for segregants (total of 81 unique pairs of barcodes per parental diploid strain).

**Translocating barcodes onto the same chromosome**. After mating, the two barcodes are located on different chromosomes (Supplementary Fig. 1E). To bring the barcodes onto the same chromosome, site-directed chromosomal translocation was induced via Cre-LoxP-mediated site-specific recombination. To do this, mated diploid colonies were pinned onto YP + 2% galactose plates and grown for 2 days. Presence of galactose induces the expression of the galactose-inducible Cre protein, causing Cre-mediated homologous recombination at the LoxP site and translocation of the chromosomes, bringing the two barcodes to close proximity. In addition to the barcodes, chromosomal translocation brings the two halves of the split URA3 marker onto the same chromosome, resulting in the reconstitution of a functional URA3 marker. Diploids that have undergone successful recombination were selected by pinning the diploid colonies onto SCM -Ura agar plates and growing for 2 days. To minimize bias due to differences in growth rate between diploids, the colonies were then pinned onto fresh SCM -Ura plates and pooled immediately with SC -Ura media. The pooled sample of ~960,000 double barcoded segregants was then spun down, re-suspended in SC -Ura media at a concentration of $2 \times 10^9$ cells/ml, and frozen in 25% glycerol for later use. In parallel, the BY/BY, 3S/3S, BY/3S, and 3S/BY parental diploids were generated in the same manner and stored at the same concentration as the segregants.

**Growth assays**. The panel of diploid segregants was evolved for 15 generations by serial batch culture under carbon limitation in 100 ml of SC -Ura media with 4% ammonium sulfate and 2% dextrose. First, 2 ml of the segregant frozen stock ($2 \times 10^9$ cells/ml) was inoculated in 198 ml of SC -Ura media and then grown in a 500 ml flask at 30 °C with 300 rpm shaking for 48 h. At saturation, the cell concentration was $1.5 \times 10^8$ cells/ml for a total of $3 \times 10^{10}$ cells, or ~$3 \times 10^4$ cells per double barcode. In parallel, 10 μl of the parental frozen stocks were inoculated in 990 μl of SCM-Ura media and grown at 30 °C with 300 rpm for 48 h. The parental diploids were then spiked into the segregant culture at a concentration of $10^{-5}$. This mixed culture served as the seed culture (time point 0 or T0) for subsequent growth time points.

Next, one-eighth or 12.5 ml of the T0 culture (~$3.75 \times 10^3$ cells per double barcode at bottleneck) was transferred into eight 500 ml flasks, each containing 87.5 ml of SC -Ura media supplemented with different drugs or chemicals (Supplementary Table 1). Two flasks contained no supplement (Glucose 1 and Glucose 2) and served as growth replicates for the experiment. The remaining T0 culture was spun down at 3000 rpm and frozen down for subsequent DNA library preparation. Each culture was grown in serial batch conditions for 15 generations, bottlenecking every ~3 generations. For each transfer, cultures were grown at 30°C with 300 rpm shaking for 48 h until saturation. One-eighth (12.5 ml) of the culture was then transferred into new 500 ml flasks, each containing 87.5 ml of the appropriate media. The remaining 87.5 ml of culture was frozen down for subsequent DNA library preparation. At each transfer, the number of cells was counted using a hemocytometer. Contamination checks for bacteria or other non-yeast microbes were performed regularly by observing the cultures under a microscope.

**Library preparation**. Frozen cultures were thawed and DNA was extracted for library preparation using the MasterPure Yeast DNA Purification kit. Any remaining RNA was removed by adding RNaseA (10 mg/ml) and incubating the sample at 37 °C for one hour. DNA was then cleaned by adding one volume of phenol:chloroform:isoamyl alcohol (25:24:1). The sample was gently mixed using a tube rotator at 30 rpm for 10 min and then centrifuged at $16,000 \times g$ for 10 min. The upper aqueous layer was transferred to a new tube and cleaned of phenol by adding one volume of chloroform:isoamyl alcohol (24:1). The sample was again mixed at 30 rpm for 10 min, centrifuged at $16,000 \times g$ for 10 min, and the upper aqueous layer was transferred to a new tube. Finally, to remove residual chloroform, the DNA sample was ethanol precipitated using one-tenth volume of 3 M sodium acetate (pH 5.5) and 2.5 volume of 100% ethanol. The resulting DNA pellet was washed with ice-cold 70% ethanol twice, and dissolved in 10 mM Tris-HCl pH 8.0. After the DNA was extracted and cleaned, a two-step PCR was used to amplify

the double barcodes for sequencing. Because only a small fraction of the total genome contains relevant information (~250 bp amplicon out of a 12 Mb genome size), we amplified 30 µg of template per time point sample, which corresponds to ~$2 \times 10^9$ genomes or ~2000 copies per double barcode.

First, a 4-cycle PCR with OneTaq polymerase (New England Biolabs) was performed in 60 wells of 96-well PCR plates, with ~500 ng of DNA template, 4.5 mM MgCl₂, 3% DMSO, and 125 µL total volume per well. Primers for this reaction were: AATGATACGGCGACCACCGAGATCTACACNXXXXXNNACA CTCTTTCCCTACACGACGCTCTT and CAAGCAGAAGACGGCATACGAGA TNXXXXXNNGTGACTGGAGTTCAGACGTGTGCTCTTCCGATCT.

The Ns in these sequences correspond to any random nucleotide and are used as unique molecular identifiers (UMIs) in downstream analysis to identify PCR duplicates. The Xs correspond to one of several multiplexing tags, which were used to distinguish different samples when loaded on the same sequencing flow cell. Multiplexing tags were designed to have a Levenshtein distance of 3 from each other, such that reads with 1 or less sequencing errors in the multiplexing tag can be assigned to the correct sample.

After amplification, 16 wells of PCR products were pooled (4 pools total), run through a NucleoSpin PCR cleanup column, and eluted in 120 µL of elution buffer. The pooled PCR products were then cleaned for the second time to remove any residual primers using a NucleoSpin column and eluted in 120 µL of elution buffer. A second 22-cycle PCR was performed with PrimeStar HS DNA polymerase (Takara) in 16 reaction tubes, with 30 µl of cleaned product from the first PCR as template and 125 µL total volume per tube. Illumina primers P1 and P2 were used for this reaction.

PCR products from all reaction tubes were concentrated into 100 µl using ethanol precipitation. The appropriate PCR band (264 bp) was isolated by agarose gel electrophoresis and quantified by Bioanalyzer (Agilent) and Qubit fluorometer (Life Technologies).

**Barcode sequencing**. Sequencing (150 bp paired-end) was performed on an Illumina HiSeq 4000 or NovaSeq 6000. Each flow cell contained at least 2 multiplexed time points and 25% genomic DNA or PhiX. Genomic DNA/PhiX was included to increase the read complexity for proper calibration of the instrument since most of the bases in each barcode read are fixed. Sequencing reads were analyzed using custom written code in Python and R, which are available on GitHub. Reads were sorted by their multiplexing tags (the Xs in the primers above) and removed if they failed to pass either of two quality filters: 1) The average Illumina quality score for the double barcode insert region must be greater than 30, and 2) the double barcode region must contain the fixed sequence ATAACTTCGTATAATGTATGCTAT with less than 2 mismatches. After filtering, MATa and MATα barcodes were extracted from the sequencing reads and fused into one double barcode.

We next used Bartender to cluster similar sequences into consensus double barcodes, with the maximum allowable sequence distance (-d) set to 2[30]. One source of bias in these counts that we wanted to avoid is PCR duplicates or other non-linearities between the amount of template for a double barcode and the number of sequences observed for that double barcode. We removed these errors by using UMIs (the Ns in the primers above). Specifically, 2 random 3-mers were attached to each template DNA molecule in the first few rounds of PCR (see section 'Library preparation' for more detail). Because the total sequence space of 2 random 3-mers (4^6 = 4,096 possibilities) is much larger than the target coverage for each double barcode (~100×), it is unlikely that any two template DNA molecules from the same time point that contain the same double barcode will be attached to identical pairs of 3-mers. Thus, sequence reads for a double barcode that contained the same pair of 3-mers were counted as PCR duplicates and removed from our final counts.

After the double barcode clusters were counted and filtered, only double barcode clusters carrying the previously determined MATa and MATα segregant and parental barcodes were extracted. All double barcodes with an average read count of <5 across the 4–5 time points were removed, as accurately estimating fitness for samples with very low read counts is difficult (Supplementary Fig. 4)[31]. In total, ~227,999 unique genotypes with an average of ~2.82 barcode replicates per genotype were detected across conditions (Supplementary Table 4). These numbers are lower than the theoretical number of double barcodes, 240,000 diploids × 4 biological replicates = 960,000. This discrepancy may be due to segregants failing to mate during the generation of diploid segregants or being lost during rearraying procedures.

**Counting double barcode reads**. Previous studies have shown that PCR amplification of DNA sequences containing two variable regions (for this study the MATa and MATα barcodes) separated by a fixed region can lead to formation of undesired chimeric molecules due to template switching[28,29]. This can result in erroneous double barcode counts, leading to errors in fitness estimation.

To identify PCR chimeras in each sample, we pooled data from all 4–5 time points and counted the number of reads that contain each combination of a parental barcode and an haploid segregant barcode. Because the parental strains were never mated to the haploid segregants, these parental-segregant double barcodes must be due to a PCR chimera. We next counted the number of times each single barcode is present in the entire data set regardless of the barcode it is paired with. We then fit a linear model between the count of each PCR chimera

and the product of the total counts of each single barcode within the pool: *# of copies of PCR chimera ~ # of copies of barcode1 x # of copies of barcode2*. Because template switching occurs randomly, we expect to see a linear relationship between the number of PCR chimeras and the abundance of the involved barcodes (average $R^2 \approx 0.623$ across environments) (Supplementary Table 5). Using this linear model, the expected number of PCR chimeras for each double barcode was calculated. The expected number of PCR chimeras were subtracted from the actual read counts to calculate the corrected counts. We estimate that the PCR chimera rate was 0.11% across environments (Supplementary Table 5). Corrected counts less than 0 were set to 0. All double barcodes with an average corrected read count of <5 across the 4–5 time points were removed.

**Fitness estimation**. The corrected read counts from the 4–5 time points were used to estimate fitness for each double barcode using a maximum likelihood algorithm PyFitSeq using default settings[31]. Fitness estimates with low maximum likelihood scores were removed, as these fitness estimates are most likely technical artifacts. Outliers with low maximum likelihood scores were defined as data scores that are more than 1.5 interquartile range below the first quartile. Additionally, for strains with three or more replicates, outlier replicates with significantly different fitness estimates were removed. Outlier replicates were determined by examining all pairwise differences in fitness estimates between barcode replicates for the same strain. Based on this distribution, outlier replicates were defined as replicates whose fitness is more than 0.5 fitness units different from the other replicates. For strains with only two replicates, if the fitness estimate difference between the two replicates was greater than 0.5, both replicates were removed. Finally, all strains with only 1 replicate were omitted from further analysis as we have no way to assess the quality of the fitness estimate. Overall, fitness estimates for ~548,046 (~190,604 genotypes with average of ~2.88 replicates per genotype) double barcodes per environment were used for the remaining downstream analysis (Supplementary Table 2).

**Heritability estimates**. Broad-sense heritability was estimated using the reproducibility of phenotype across replicates. The linear model, *fitness ~ genotype* was used, where *genotype* is a categorical value corresponding to the 2–4 biological replicates for each diploid segregant. Broad-sense heritability was calculated by taking the sum of squares of genotype and dividing it by the total sum of squares. Because large memory overhead is required to calculate broad-sense heritability for the entire dataset, we calculated 1,000 broad-sense heritability estimates for a smaller subset of randomly selected 2500 genotypes. The reported broad-sense heritability and standard errors are the mean value and the standard error across the 1000 tests.

Variances explained by additive, dominance, and epistasis effects were estimated using the 'sommer' package in R[59]. The additive, dominance, and epistasis relationship matrices were calculated using the A.mat(), D.mat(), and E.mat() functions, respectively. Variances explained by additive, dominance, and epistasis were then estimated by dividing the variance of the respective components by the total variance. Similar to the broad-sense heritability estimates, we calculated 1,000 variance estimates for a smaller subset of randomly selected 2500 genotypes. The reported variance estimates and standard errors are the mean value and the standard error across the 1000 tests.

**Quantile normalization of fitness**. For each genotype in each condition, the average fitness estimate across all barcode replicates was calculated. Because the distribution of the average fitness estimates was slightly left skewed, average fitness estimates were quantile normalized such that the data is normally distributed using the 'bestNormalize' R package (Supplementary Fig. 6)[60]. This quantile normalized fitness estimate was used as the *fitness* phenotype for all downstream analysis.

**Genome-wide scan for one-locus effects**. Our experimental design resulted in genotypes that are not equally related to each other. For example, two diploid segregants that share a parental haploid segregant are more related to each other than two diploids that do not share a parental haploid segregant. We found that differences in relatedness had a large effect on the fitness of the diploid segregants. Thus, failing to account for family structure could inflate type I error and result in false positives. To account for these potential errors, we detected one-locus effect loci using FaST-LMM[37,38], which runs a mixed effects linear model (MLM) with a spectrally decomposed genetic similarity matrix (GSM).

We used FaST-LMM's single_snp function to perform a genome-wide linear mixed effect analysis. The genotype table was reformatted as a binary biallelic genotype table (BED) prior to running FaST-LMM using PLINK version 1.07[61]. To avoid proximal contamination, all SNP markers that are located on the same chromosome as the SNP marker that is being tested were omitted from the calculation of the GSM by setting the *leave_out_one_chrom* function as TRUE. To determine appropriate significance thresholds, 1000 permutations were conducted with the correspondence between genotypes and phenotypes randomly shuffled each time. Among the minimum *p* values obtained in the permutations, the fifth quantile was identified and used as the threshold for determining significant loci.

To detect loci that are closely linked to a nearby locus, we ran multiple rounds of stepwise forward regression in FaST-LMM. For the second round of forward regression, we included the most significant SNP marker from each chromosome

that was above the significance threshold from the initial genome-wide scan as a covariate in the mixed effect linear model. In the third round, all SNP markers detected in the first two rounds were added as covariates in the mixed effect model. This process was repeated until no significant SNP markers were detected. The confidence interval of a locus was defined as the region surrounding a peak marker in which the significance of detection was within three $-\log10(p\text{-value})$ of the maximal significance observed at the peak position (hereafter referred to as "3-LOD drop").

**Family-level scans for one-locus effects**. We also performed scans for one-locus effects within families generated from the same *MATa* haploid parental strain. Results from families generated from the same *MATα* haploid parental stains were not used as the difference in sample size (~600 diploids in *MATa* families vs ~400 diploids in *MATα* families) had a huge impact on the statistical power to detect loci. Because all genetic differences between diploids within a family are random, they do not need to be corrected for family structure prior to mapping. Genome-wide mapping was conducted in each family using a fixed effects linear model in R using the lm() function: *lm(fitness ~ locus)*. The *fitness* term corresponded to the vector of quantile normalized fitness values of individuals within a family and the *locus* term corresponded to the vector of these individuals' genotypes at a given marker. Because diploids within a family are derived from the same *MATa* haploid, they possess only two possible genotype states: BY/BY and BY/3S or BY/3S and 3S/3S, depending on the allele present in the *MATa* progenitor. Although the *locus* term was encoded as a categorical variable here, the coding of the *locus* term should not matter. The categorical coding should be equivalent to a count of an allele because either defines two categories. The *p*-value of the *locus* term was obtained using the summary.aov() in R. Among markers exceeding a significance threshold, the most significant marker from each chromosome was identified and utilized in subsequent rounds of forward regression. Specifically, identified loci were included in the fixed effects linear model as a covariate, e.g. *fitness ~ known_locus1 + known_locus2 + … known_locusN + locus*. This process was repeated until no additional *locus* terms were discovered within an environment. The significance threshold for these forward scans was established by pooling the *p* values from 1000 permutations of 10 randomly selected families. In each permutation, the correspondence between genotype and phenotype was shuffled within a family. From the resulting distribution of minimum *p*-values obtained in each permutation, the fifth quantile was identified and used as the significance threshold. After calling peaks, loci detected in multiple families were consolidated using 3-LOD drops within each family.

**Identification of loci enriched for detections across families**. We identified distinct loci that were enriched for detections across families[39]. We divided the genome into nonoverlapping 20 kb bins and determined the maximum number of detections expected by chance at a Bonferroni-corrected $p = 0.05$/(the number of 20 kb nonoverlapping bins). The threshold was then obtained from a Poisson distribution with $\lambda =$ (# of detections across all families within the environment)/(the number of 20 kb nonoverlapping bins tiling the genome) using the qpois() function in R, with lower.tail set to FALSE.

**Removing additive effects and correcting for family structure**. For detection of dominance effects and interactions, family structure was corrected using a previously described strategy[23]. Specifically, we first estimated the fitnesses of each parental haploid segregant by calculating the mean fitness of all diploids that originated from each parental strain (additive genetic background contribution to fitness). Next, the expected midparent value for each diploid was obtained by taking the average of the 2 parental fitnesses. If phenotypic variation is due to only additive effects, then we expect the fitness of a diploid to be the same as its midparent value. Deviations from the midparent value were then determined by fitting a linear model between fitness and the midparent value, *fitness ~ midparent value*, and taking the residuals (hereafter referred to as 'residuals'). Not only does this correct out the additive portion of each diploid's phenotype, it also effectively accounts for family structure (Supplementary Fig. 10)[23]. These *residuals*, or the non-additive portions of each diploid's phenotype, were used as phenotype for the subsequent scans for loci involved in non-additivity, such as dominance and genetic interactions.

**Genome-wide scans for loci with dominance**. To detect loci with one-locus non-additive effects (e.g. dominance), we ran multiple rounds of forward regression. In the first round, a genome-wide scan was conducted using the following fixed effects linear model: *residuals ~ locus*, where *locus* is the genotype of the diploids at a given SNP marker encoded as a categorical variable. Here, the significance of the *locus* term was tested. The significance threshold was determined by conducting 1000 permutations with the correspondence between genotypes and *residuals* shuffled each time. Among the minimum *p* values obtained in the permutations, the fifth quantile was identified and used as the threshold for determining significant loci. The most significant SNP marker from each chromosome that was above the significance threshold was identified as a significant locus. For subsequent rounds of forward regression, identified loci were included in the fixed effects linear model as a covariate, e.g. *residuals ~ known_locus1 + known_locus2 + … known_locusN + locus*. The most significant SNP

marker from each chromosome that was above the significance threshold was again identified as a significant locus. This process was repeated until no additional *locus* terms were discovered for an environment.

**Degrees of dominance**. For each locus detected in the scan for dominance effects, the magnitude of dominance was calculated in the following manner. First, the data was subsetted based on the genotype state at the focal locus (e.g., BY/BY, BY/3S, 3S/3S). Then, the mean *fitness* value for each genotype state was calculated. Additive effect sizes of the 3S allele were calculated by subtracting the mean *fitness* of diploids that are BY/BY at the focal locus from the mean *fitness* of diploids that are 3 S/3 S at the focal locus and dividing by two. Dominance effect sizes for each focal locus were calculated by subtracting the mean *fitness* of diploids that are BY/3S at the focal locus from the average of the mean *fitness* of diploids that are BY/BY and 3S/3S at the focal locus (i.e., the additive expectation for heterozygotes). Dominance effect sizes were then normalized based on the absolute additive effect sizes of the 3S allele relative to the BY allele at the same loci. Normalized dominance values ranging from $-0.9$ to 0 were classified as incomplete dominance for the deleterious allele, while values ranging from 0 to 0.9 were classified as incomplete dominance for the allele conferring higher fitness. Values that were between $-0.9$ to $-1.1$ or 0.9 to 1.1 were classified as complete dominance for the deleterious and fit allele, respectively. Values $< -1.1$ or $>1.1$ were classified as underdominant or overdominant, respectively.

**Comprehensive scan for pairwise interactions**. We conducted a genome-wide scan for pairwise interactions using the following linear fixed effect model: *residuals ~ locus1 + locus2 + locus1:locus2*. Here, the significance of the *locus1:locus2* interaction term was tested. Simpler terms were included in each model to ensure that variances due to one-locus non-additive effects (e.g., dominance) were not erroneously attributed to more complex terms. All *locus* terms were treated as categorical values, including the *locus1:locus2* interaction term, such that all 9 genotype classes (BY/BY-BY/BY, BY/BY-BY/3S, BY/BY-3S/3S, BY/3S-BY/BY, BY/3S-BY/3S, BY/3S-3S/3S, 3 S/3S-BY/BY, 3S/3S-BY/3 S, 3S/3S-3S/3S) are treated as independent categories rather than continuous numerical values. The significance threshold was determined by conducting 1,000 permutations with the correspondence between genotype and *residuals* shuffled each time. Among the minimum p-values obtained in each permutation, the fifth quantile was identified and used as the significance threshold for our comprehensive scan of all two-loci interactions. To reduce computational time, 10,000 random pairs of loci were randomly selected for each permutation rather than all possible pairs of loci. All significant pairs of loci where both loci were within 3-LOD drop of each other were consolidated. For each set of overlapping loci, the SNP marker with the most significant p-value was used for downstream analysis while less significant markers were recorded but treated as the same genetic effect and not used downstream.

During our comprehensive scan for pairwise interactions, we detected several hubs that were involved in large numbers of interactions. These interactions often spanned the entire length of the genome, making it difficult to differentiate sites interacting with these hubs. In such cases, a forward regression approach was conducted. Specifically, we first ran the same fixed effects linear model as the comprehensive scan for pairwise interactions, but with *locus1* fixed for a hub locus, e.g., *residuals ~ hub + locus2 + hub:locus2*. The significance of the *hub:locus2* interaction term was tested. Significance thresholds were determined using the same permutation strategy as the comprehensive scan. The most significant interacting locus from each chromosome that was above the significance threshold was identified as a significant interactor of a hub locus. For subsequent rounds of forward regression, all identified interactions were included in the fixed effects linear model as covariates, including the simpler terms, e.g. *residuals ~ hub + locus2 + known_locus1 + known_locus2 + … known_locusN + hub:known_locus1 + hub:known_locus2 + … hub:known_locusN + hub:locus2*. The most significant interacting locus from each chromosome that was above the significance threshold was again identified as a significant interactor with a hub locus. This process was repeated until no additional *hub:locus2* terms were discovered for each environment. To avoid double counting interactions, all pairwise interactions identified in the comprehensive scan that involved a locus within 50 kb of a hub were removed.

**Scan for three-locus effects**. A comprehensive scan for three-locus effects using all SNP markers is computationally expensive. Instead, we scanned for three-locus effects using a smaller subset of 579 SNP markers, where each SNP marker was at least 5 cM apart. The following fixed effects linear model was used: *residuals ~ locus1 + locus2 + locus3 + locus1:locus2 + locus1:locus3 + locus2:locus3 + locus1:locus2:locus3*. Similar to the scan for pairwise interactions, all *locus* and interaction terms were treated as categorical values. Here, the significance of the *locus1:locus2:locus3* interaction term was tested. Simpler terms were included in each model to ensure that variances due to one-locus non-additive effects (e.g., dominance) and pairwise interactions were not erroneously attributed to more complex terms. As before, the significance threshold was determined by conducting 1000 permutations with the correspondence between genotype and *residuals* shuffled each time. Among the minimum *p*-values obtained in each permutation, the fifth quantile was identified and used as the significance threshold for our comprehensive scan of all two-loci interactions. Similar to the pairwise interaction

scan, for each permutation, 10,000 trios of loci were randomly selected for each permutation rather than all possible trios of loci.

**Resolving hub loci**. Hub loci were resolved using data from family-level scans in each of the *MAT**a*** families. The peak position of each hub was defined as the most significant marker for that locus across all families. Family-level detections with confidence intervals overlapping the peak position of a hub were identified. The first and last coordinates of all confidence intervals overlapping a peak position were compared, and the coordinates closest to each side of a peak position were recorded as the minimum bounds for a given hub.

**Estimating the fraction of epistatic effects that involve dominance**. To understand how a locus modifies the effect of an interacting locus in a pairwise interaction, each pairwise interaction was partitioned into four epistasis types: *additive-additive, dominance-additive, additive-dominance, and dominance-dominance*[62]. The following linear fixed effect model was used: *residuals ~ locus1 + locus2 + additive_locus1:additive_locus2(a1:a2) + dominance_locus1: additive_locus2(d1:a2) + additive_locus1:dominance_locus2(a1:d2) + dominance_ locus1:dominance_locus2(d1:d2)*. *Additive_locus* terms were treated as numerical values, while *locus* and *dominance_locus* terms were treated as categorical values. The percent variance explained (PVE) by each of the four epistatic types was then estimated by taking the sum of squares of each interaction term and dividing it by the total sum of squares. Using these PVE values for each interaction term, the fraction of epistatic effect involving dominance was then calculated. To estimate the fraction in which *locus1*'s modifying effect on *locus2* acts on dominance, the following equation was used: $(d1:a2 + d1:d2)/(a1:a2 + d1:a2 + a1:d2 + d1:d2)$. Conversely, to estimate the fraction in which locus2's modifying effect on locus1 acts on dominance, the following equation was used: $(a1:d2 + d1:d2)/ (a1:a2 + d1:a2 + a1:d2 + d1:d2)$.

**Examining how combinations of modifiers explain effect size variance of hubs**. Using the PVE values for the four epistatic types (see section 'Estimating the fraction of epistatic effect that involves dominance'), we examined how loci that interact with a hub contribute to additive and dominance effect size variance across different genetic backgrounds. For each two-locus interaction involving a hub, the magnitude in which the hub modifier affects the additive component of the hub ('add_mod') was determined by adding the PVE of a1:a2 and a1:d2 interaction terms, where *locus1* is the hub. The magnitude in which the hub modifier affects the dominance component of the hub ('dom_mod') was determined by adding the PVE of d1:a2 and d1:d2 interaction terms. Significantly large outliers in *add_mod* or *dom_mod*, defined as values more than 1.5 interquartile range above the third quartile, were identified as major effect modifiers.

To examine how modifiers collectively change the effect size of hubs, four modifiers with the largest modifying effects on additivity and dominance were chosen for each hub. Data was then subsetted into 81 genotype classes based on the genotype state at the four modifiers. For each genotype class, the data were further subsetted based on the genotype state of the focal hub locus. Additive effect sizes for each genotype class were calculated by subtracting the mean *fitness* of diploids that are BY/BY at the focal hub from the mean *fitness* of diploids that are 3S/3S at the focal hub. Dominance effect sizes for each genotype class were calculated by subtracting the mean *fitness* of diploids that are heterozygous at the focal hub from the average of the mean *fitness* of diploids that are BY/BY and 3S/3S at the focal hub.

**Reporting summary**. Further information on research design is available in the Nature Research Reporting Summary linked to this article.

## Data availability

All data generated or analyzed during this study are included in this published article (and its Supplementary information files). A Data availability statement is included in the manuscript. Raw barcode sequencing data are available from the NCBI Sequence Read Archive as accession PRJNA781980. Data used in analyses are available in Mendeley data (https://data.mendeley.com/datasets/96ghpptzvf). S288C reference genome ver R9-1-1 was used in this study and is available at http://sgd-archive.yeastgenome.org/sequence/ S288C_reference/genome_releases/.

## Code availability

All code used to analyze data, perform statistical analyses, and generate figures is available at Github (https://github.com/tmatsui2/Matsui-et-al.−2021-Supplemental-Information).

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

## Acknowledgements
The authors thank members of the Ehrenreich and Levy labs for feedback on drafts of this manuscript and Oscar Aparicio for an *HO* plasmid. The authors also thank Adam Dziulko for helping to create the panel of barcoded diploid strains. This work was funded by grants R01GM110255 and R35GM130381 from the National Institutes of Health to I.M.E., grants R01AI164530 and R01HG010378 from the National Institutes of Health to S.F.L., support from the National Institute of Standards and Technology and The Joint Initiative for Metrology in Biology to S.F.L., as well as funds from the University of Southern California and Stanford University to I.M.E. and S.F.L., respectively. M.N.M. and R.S. were partially supported by Research Enhancement Fellowships from the University of Southern California Graduate School. J.J.H. was supported by the NIGMS training grant T32GM118289.

## Author contributions
T.M., M.N.M., K.R., S.F.L., and I.M.E. conceptualized this project. T.M., M.N.M., J.J.H., and R.S. constructed and genotyped the segregants. T.M. and M.N.M. performed the experiments and analyzed the data. T.M., M.N.M., S.F.L., and I.M.E. wrote and edited the manuscript.

## Competing interests
The authors declare no competing interests.
