## [Peer Review File · Nature Communications]

Title: The interplay of additivity, dominance, and epistasis on fitness in a diploid yeast crossREVIEWER COMMENTS

Reviewer #1 (Remarks to the Author):

In this paper, Matsui et al explore the relationship between genetic interactions and phenotypic variation across environments using a diploid budding yeast model system. Most genome-scale studies of genetic interactions in yeast have focused on interactions between loss-of-function alleles in haploids, precluding a systematic assessment of dominance effects. The authors developed and applied a 'double bar-coding' strategy to enable phenotyping of a large diploid cross. The genomes of the parental strains (standard lab strain and a clinical isolate derivative) differ at ~45000 SNPs and the parents of a diploid segregant can be identified by a double barcode on one chromosome. Competitive growth assays were used to measure the relative fitness of ~200,00 diploids in seven growth conditions, revealing considerable phenotypic diversity, largely attributable to genetic factors. The experiments revealed that epistatic hubs tend to govern both additive and non-additive genetic interactions, and that dominance effects are common and variable in magnitude depending on the interacting loci alleles. These are interesting and important observations, that should be of broad interest to geneticists.

Below are some issues/questions that I think should be addressed:

[1] Figure 1D shows the correlation between replicate strains (same genotype but independently barcoded before mating) is not very good. The authors have done a lot of work with barcode fusion systems and fitness inference from barcode sequencing, so the method is solid. Thus, more discussion about the possible explanations for the low correlation would be useful. It would also be good to re-sequence a couple of the diploids strains and to try to estimate the rate of gene conversion.

[2] The statistical methods could use some attention. Since the 200,000 diploids only have 1,000 parents, 'state-of-the-art' methods for QTL mapping cannot be used. Instead, the authors use a method that can do some amount of correction for the "family" structure of the diploids. This method gives disappointing results and to explore the data, the authors map QTLs independently for each of the 400 "families" that have one parent fixed and 600 other parents. By inferring QTLs 400 times, the authors are making multiple comparisons that go uncorrected (or, at least, there is no mention of correction for multiple comparisons) and this likely means that many of the significant QTLs are not, in fact, significant. This seems likely since the method that uses the full data set calls many fewer QTLs. To explain the discrepancy, the authors suggest that it is possible that the correction for the "family" structure is overly conservative and that the adhoc method is outputting useful inferences. It may well be that the FaST-LMM method deals with this issue, but it would still be good for the authors to include a more thorough defense of the method in the text.

[3] The rest of the manuscript uses the inferences discussed above. The analysis is interesting, but it would be useful to compare to the same analysis on the more conservative set of QTLs.

[4] I think that figure 4C is confusing since the x-axes are lined up with the other panels even though they are completely different dimensions.

Reviewer #2 (Remarks to the Author):

Key results:

The authors report an exciting high-throughput advanced intercross experiment in diploid yeast to identify genetic variation that regulates fitness in various environments. Because the cross is so high-throughput and the mapping population comprises diploid outbred genomes, the authors are able to identify and characterize non-additive genetic effects (ie, dominance and two-locus and three-locus epistasis). They decompose epistatic loci based on how they modify the additive and dominance effects of their interacting loci. They identify key hub loci that have many epistatic interactions and show trends in how different types of epistatic loci modify effects differently. They identify the mating locus as a dominance modifier, which matches finding in plants at loci with unique allelic series.

Validity:

The authors' interpretation of the data and conclusions are valid and appropriate. They caveat that their findings are specific to yeast, but one can reasonably then speculate about more genetically diverse natural yeast populations and even multicellular organisms.

Significance:

The findings of this study are highly significant to the field of complex trait genetics. It is incredibly challenging to design an experimental cross to assess wide-scale dominance and epistasis, in that the genomes must be largely outbred and the sample sizes need to be very large. The authors have designed a sophisticated, novel experiment to create just such a large outbred diploid yeast cross, leveraging the ability of yeast models to generate very large numbers of genetically unique individuals (as well as replicates!). This is a very well-designed experiment for detecting and characterizing non-additive genetic effects.

Data and methodology:

As mentioned above in the Significance section, the experimental cross was very well-designed. They assess the quality of the data (and potential replicability) by also including biological replicates (identical diploid genomes with different double-barcodes) and a replicate experiment of one of the environments (glucose), finding the fitness phenotype is correlated across both types of replicates.

I briefly viewed the supplemental data files, which seemed like appropriate summary files to include. I also looked at the GitHub repo, which looked to have all the described components. The authors could consider putting a frozen version of the repo on a repository like figshare, which is potentially superior in terms of reproducibility because it can't be deleted by the author.

Analytical approach:

The statistical analyses were valid and robust. When mapping across all the individual yeast genomes, they used a linear mixed effect model (LMM) to account for unequal relatedness of diploid genomes, essentially representing population structure across the families. They even confirmed the issue of structure by showing the extreme inflation for p-values when population structure is ignored when

mapping across families. They also mapped QTL at the level of families, at which point the diploid genomes are equally related (in expectation) and share one parent, at which point the LMM is theoretically unnecessary. Furthermore, the approaches for measuring heritability and partitioning genetic effects into additive, dominance, and epistasis were appropriate.

Suggested improvements:

I don't think any major changes or improvements are necessary for this manuscript. I do have some broad comments, which could be used for minor adjustments to sections of the manuscript.

- Some readers may appreciate some comparison of genetic loci and their effects across environments, which could identify some interesting biology (eg, does the regulating genetic variation make biological sense in terms of potential causal genes and environments?). However, this doesn't really match the main questions tackled in this manuscript and could thus distract from the message on non-additive genetic effects.

- LMM vs fixed effects regression:

- o The authors note that the across-family analysis through LMM didn't yield more loci as might be expected given such a large increase in sample size compared to family-specific scans. This result doesn't actually surprise me that much given the complexity of the overall population structure. The genetic relationship matrix will encompass coarser population structure, ie, the family blocks of genomes that share an entire parental haplotype, as well as finer-scale differences like the chance differences in shared descent within a family. It's actually very impressive that the LMM does such a good job of reducing the potential for false positives due to this population structure (Fig S7).

- o Furthermore, given how effects can vary across genetic backgrounds, which the authors demonstrate nicely, it's not surprising that fewer loci are detected across the whole population. The family-specific scans can be viewed as testing a family-QTL interaction term, so if genetic effects vary across families, it makes sense that a lot would be missed by expecting a consistent QTL effect. There are some QTL detected across families with LMM that don't show up in the family-specific scans (Fig 2A-B). I would speculate that those are potentially weaker QTL that also happen to be fairly consistent across families.

- o There is some support for using LMMs even when individuals are expected to be equally related (genomes are essentially randomized with respect to each other), in order to account for the observed imbalance in relationships (PMID: 24906874). I think the impact on power is fairly minimal in this situation, and fixed effects regression is justifiable over LMMs at times, such as due to the computational burden when scanning many families or the overall population is very large.

- The large-scale adjustment for additive effects was very interesting, and not something I was familiar with based on my work largely in inbred mice. Showing the examples (Figs 2F, 3D) very nicely demonstrates its effectiveness for identifying non-additivity.

- Multiple testing correction:

- o The authors describe controlling for the FWER through permutations within a scan-type. Technically there is additional multiple testing occurring across the different analysis types. I think their approach is acceptable though given the goal of looking at non-additive genetic effects. A complex multi-step correction for multiple testing would just distract from this work.

- Chromosome IV is a distinct outlier in terms of founder allele balance (Fig S2). Is there anything known about that region in recombinant yeast populations? Maybe it's specific to these parental strains. It does

distract from the core message, so I think it's reasonable to not dwell on it.

Clarity and context:

The manuscript is well-written and I believe easy to follow for a reader with a background in genetics. The Methods section is also nicely written, and covers all the different analyses.

- I wonder if it would be good to add information about the phenotype to the manuscript title? Maybe adjust it to be "The interplay of additivity, dominance, and epistasis on fitness in a diploid yeast cross". I think the findings can be interpreted broadly across phenotypes, so I'm not sure it needs to be adjusted.
- The authors mention consolidating loci by using the 95% confidence interval (CI) on QTL position. They don't mention what this interval is based on. CIs for QTL position are usually approximate, using a support interval or parametric bootstrap. I'm sure whatever they used is fine, but it could be good to report it, even if it were the default in Fast-LMM or something like that.
- Fig 2E could use more description. The authors show that loci effects detected across families are clearly larger on average than those detected only at the family-level. I think that is very clear, and makes sense, but it's not clear to me what the factor levels are on x-axis. Maybe they've been sorted into bins to show the full spectrum of effects? I think just a little more detail in the captions would suffice.

• Minor comments

o I would slightly rephrase this sentence in the 2nd paragraph of the first section. "Most empirical studies of non-additive genetic effects have focused on haploid or inbred individuals (14–16), which provide higher statistical power to detect loci due to their nominal levels of heterozygosity." "Nominal" has dual meanings, in that it could be referring to a stated level of a summary (ie, nominal p-value vs empirical p-value), whereas I assume the authors mean it as very low levels of heterozygosity. I would consider rephrasing with "minimal" or even "lack of heterozygosity", since residual heterozygosity in inbreds is usually low.

o The words "strain", "genotype", and "diploid" are used as synonyms throughout the manuscript, which is correct given the context, meaning a unique diploid genome. But possibly some readers will be confused or read "genotype" and think specifically of a locus. Just something to consider.

o I would consider expanding the title of the Methods section "Correcting for family structure" to something like "Removing additive effects and correcting for family structure".

o Technically the coding of the locus effect in Methods section "Family-level scans for one-locus effects" shouldn't matter; the categorical coding should be equivalent to a count an allele, because either defines two categories. If both homozygotes are present (as in the cross-family analysis), then the coding should matter and can distinguish an additive term. The authors do correctly describe the variable codings for distinguishing additive and non-additive effects in the more complicated model in Methods section "Estimating the fraction of epistatic effect that involves dominance".

• Typos

o The word "strains" is repeated at the end of the first sentence of the second paragraph.

o Typo in last sentence of the first paragraph of the Methods section "Counting double barcode reads": "This can result in erroneous double barcode counts and result errors in fitness estimation." Could re-write as: "This can result in erroneous double barcode codes, leading to errors in fitness estimation."

o Typo in the second paragraph of the Methods section "Counting double barcode reads": "We then fit a

linear model between the count of each PCR chimera and the product of these total count of each single barcode within the pool:”. Could re-write as: “We then fit a linear model between the count of each PCR chimera and the product of the total counts of each single barcode within the pool:”

References:

The authors could add some additional citations for family/population structure and linear mixed effects models for genetic mapping (first paragraph for “Mapping within interrelated families increases statistical power” section). For example, Kang et al 2008 (EMMA), Kang et al 2009 (EMMAX), and Zhou & Stephens 2012 (GEMMA). This would provide greater background for an area that saw a lot of development roughly 10 years ago. The manuscript also isn’t over-stuffed with citations, so a few more would seem fine.

There were also some missing references in the Methods:

- Missing reference in the second sentence of the Methods section “Whole-genome sequencing of parental haploid segregants”.
- Missing reference in the first sentence of the second paragraph of the Methods section “Barcode sequencing”.
- Missing reference in the second sentence of the third paragraph of the Methods section “Barcode sequencing”.
- Missing reference in the Methods section “Estimating the fraction of epistatic effect that involves dominance”.

Final comment:

I really enjoyed reviewing this manuscript and believe it represents an impressive contribution to the field of complex traits genetics. I recommend its acceptance and publication.

November 30, 2021

To the Reviewers,

Thank you for your thoughtful assessment of our manuscript. In response to your comments, we have modified the manuscript, as discussed in the point-by-point below. Our responses are shown in red. We hope that you will find this revision suitable for publication in Nature Communications.

Sincerely,

Takeshi Matsui, Martin Mullis, Sasha Levy, and Ian Ehrenreich

Reviewer #1 (Remarks to the Author):

In this paper, Matsui et al explore the relationship between genetic interactions and phenotypic variation across environments using a diploid budding yeast model system. Most genome-scale studies of genetic interactions in yeast have focused on interactions between loss-of-function alleles in haploids, precluding a systematic assessment of dominance effects. The authors developed and applied a 'double bar-coding' strategy to enable phenotyping of a large diploid cross. The genomes of the parental strains (standard lab strain and a clinical isolate derivative) differ at ~45000 SNPs and the parents of a diploid segregant can be identified by a double barcode on one chromosome. Competitive growth assays were used to measure the relative fitness of ~200,00 diploids in seven growth conditions, revealing considerable phenotypic diversity, largely attributable to genetic factors. The experiments revealed that epistatic hubs tend to govern both additive and non-additive genetic interactions, and that dominance effects are common and variable in magnitude depending on the interacting loci alleles. These are interesting and important observations, that should be of broad interest to geneticists.

Thank you for this positive feedback.

Below are some issues/questions that I think should be addressed:

[1] Figure 1D shows the correlation between replicate strains (same genotype but independently barcoded before mating) is not very good. The authors have done a lot of work with barcode fusion systems and fitness inference from barcode sequencing, so the method is solid. Thus, more discussion about the possible explanations for the low correlation would be useful. It would also be good to re-sequence a couple of the diploids strains and to try to estimate the rate of gene conversion.

We did indeed find a difference in the correlation between the same strain grown in replicate flasks ($\rho = 0.86$) and replicate strains grown within the same flask ($\rho = 0.63$). We first note

that even $\rho = 0.63$ is a very strong, highly significant correlation ($p < 2.2 \times 10^{-16}$). As the reviewer suggests, one potential explanation is that this difference is caused by genetic differences between replicate strains (LOH, mutation, aneuploidy). However, the explanation for the difference between these two correlation measures is unlikely to be biological. Rather, the major difference is that the initial frequency of each strain within the pool is approximately equal in the replicate growth experiment but highly variable between replicates in the same growth experiment. We have previously shown that the accuracy of fitness estimates (measurement noise) is significantly impacted by the initial frequency of a strain (Li et al. 2018, Cell Systems 7(5) 521-525). To determine if this was the major factor causing the difference between these two correlation measures, we examined how the correlation between replicate strains changes if we only include replicate strains that are both abundant in the pool. As we increased the required abundance threshold (double barcode count at the first time point), we found that the correlation improved to even exceed that of the replicate growth correlation, suggesting that initial frequency is the main cause of the difference. This analysis is shown in a new Figure S4B and below.

To better explain this phenomenon to the readers, we have added the following text to the 4th paragraph under section “Phenotyping of a large diploid cross by barcode sequencing” in the main text:

“Noise between biological replicates was primarily caused by one or more replicates being present at a low abundance within the pool, resulting in a less accurate fitness estimate (**Fig. S4**).”

We also added the following text to the 5th paragraph of the same section to explain how our heritability estimates are in line with previous studies that use colony size as a phenotype :

“These heritabilities estimates are similar to other yeast studies in which fitness was measured based on colony area size”

We would also note that all analyses presented in this manuscript were performed using the average of at least two replicate fitness estimates (strains) for each genotype in order to minimize the effect of measurement noise. For this reason, we do not believe that the measurement noise noted by Reviewer 1 had a material impact on our study.

[2] The statistical methods could use some attention. Since the 200,000 diploids only have 1,000 parents, 'state-of-the-art' methods for QTL mapping cannot be used. Instead, the authors use a method that can do some amount of correction for the "family" structure of the diploids. This method gives disappointing results and to explore the data, the authors map QTLs independently for each of the 400 "families" that have one parent fixed and 600 other parents. By inferring QTLs 400 times, the authors are making multiple comparisons that go uncorrected (or, at least, there is no mention of correction for multiple comparisons) and this likely means that many of the significant QTLs are not, in fact, significant. This seems likely since the method that uses the full data set calls many fewer QTLs. To explain the discrepancy, the authors suggest that it is possible that the correction for the "family" structure is overly conservative and that the adhoc method is outputting useful inferences. It may well be that the FaST-LMM method deals with this issue, but it would still be good for the authors to include a more thorough defense of the method in the text.

We agree with Reviewer 1 that controlling for multiple testing in the family-level genetic mapping is important and, as discussed in the Methods, we attempted to do so in our initial submission using a permutation-based strategy. With this said, we acknowledge that perhaps a more conservative thresholding approach could have value. The challenge is that the most appropriate way to control for multiple testing in this analysis is unclear for both biological and technical reasons. Biologically, as we have shown, genetic background plays a major role in modifying the effects of the same loci across families. This is true even for some loci detected using FaST-LMM and the full data. Also, technically, mapping within the families was performed using forward regressions, in which tests in distinct families typically had different cofactors in the second and later steps of a within-family scan. In addition to tests potentially involving different models, we encountered another technical challenge: many other conceivable permutation strategies were computationally prohibitive because of the large dataset and

complex family structure. For these reasons, we attempted to address the spirit of Reviewer 1's concerns, which we interpreted as meaning that both the family-level scans may have been overly liberal and the advantage of using family-scans could have been overstated.

We made two changes to the paper to address these issues. First, we attempted to use data from multiple families to obtain stronger support for loci detected by the family-level scans. True loci detected within families should have a greater chance of being detected in multiple families than false positive loci. Taking advantage of this property, we identified distinct loci that were identified in a greater number of families than expected by chance using a previously described approach (to our knowledge first reported in Brem et al. 2002. Science). Specifically, if detections are random, they will be uniformly distributed throughout the genome. Distinct loci departing from this expectation can be identified using a Poisson-based threshold accounting for the number of nonoverlapping windows in a genome and the total number of detections across all families in a given environment. When we did this, we found ~22 'enriched' loci per environment; these loci are unlikely to be false positives. Many of these enriched loci that are supported by detections in a significant number of families were not identified by FaST-LMM and similarly many FaST-LMM loci were not detected as enriched across families. Our interpretation of these results is that the family-level scans and FaST-LMM are complementary, detecting sets of loci that are only partially overlapping. Presumably, as discussed below and now in the paper, the methods differ in their abilities to handle the various complexities of our dataset.

Second, we modified the section on mapping one-locus effects so that the results and conclusions are presented in a more balanced way. We eliminated the 'provides higher statistical power' part of the section title. Also, we rewrote the final part of this section, simply summarizing distinct results from the two mapping approaches. We believe this material will remain interesting to many geneticists because it speaks to some of the challenges in comprehensively mapping loci in populations with complex genealogical structure and genetic architectures. Because these issues are not the main focus of the manuscript (as stated by Reviewer 2) and also because these matters have little bearing on later parts of the paper (as discussed in the response to the next comment from Reviewer 1), we believe the modifications we have made are sufficient to address Reviewer 1's concerns.

[3] The rest of the manuscript uses the inferences discussed above. The analysis is interesting, but it would be useful to compare to the same analysis on the more conservative set of QTLs.

The only time the data was analyzed as individual families was for detection of one-locus effects. All subsequent linkage mapping for loci with dominance effects, two-locus effects, and three-locus effects were conducted in an unbiased manner using the entire dataset and were administered independently of the results found from the family-level scans. To correct for multiple testing in these scans, 1,000 permutations were conducted with the correspondence between genotypes and phenotypes randomly shuffled each time. Among the minimum p values obtained in the permutations, the fifth quantile was identified and used as the threshold for

determining significant loci. Only loci identified in these conservative scans are included in Figure 3.

[4] I think that figure 4C is confusing since the x-axes are lined up with the other panels even though they are completely different dimensions.

Thank you for this comment. We have changed the layout of 4C.

Reviewer #2 (Remarks to the Author):

Key results:

The authors report an exciting high-throughput advanced intercross experiment in diploid yeast to identify genetic variation that regulates fitness in various environments. Because the cross is so high-throughput and the mapping population comprises diploid outbred genomes, the authors are able to identify and characterize non-additive genetic effects (ie, dominance and two-locus and three-locus epistasis). They decompose epistatic loci based on how they modify the additive and dominance effects of their interacting loci. They identify key hub loci that have many epistatic interactions and show trends in how different types of epistatic loci modify effects differently. They identify the mating locus as a dominance modifier, which matches finding in plants at loci with unique allelic series.

Validity:

The authors' interpretation of the data and conclusions are valid and appropriate. They caveat that their findings are specific to yeast, but one can reasonably then speculate about more genetically diverse natural yeast populations and even multicellular organisms.

Significance:

The findings of this study are highly significant to the field of complex trait genetics. It is incredibly challenging to design an experimental cross to assess wide-scale dominance and epistasis, in that the genomes must be largely outbred and the sample sizes need to be very large. The authors have designed a sophisticated, novel experiment to create just such a large outbred diploid yeast cross, leveraging the ability of yeast models to generate very large numbers of genetically unique individuals (as well as replicates!). This is a very well-designed experiment for detecting and characterizing non-additive genetic effects.

Data and methodology:

As mentioned above in the Significance section, the experimental cross was very well-designed. They assess the quality of the data (and potential replicability) by also including biological replicates (identical diploid genomes with different double-barcodes) and a replicate experiment of one of the environments (glucose), finding the fitness phenotype is correlated across both types of replicates.

I briefly viewed the supplemental data files, which seemed like appropriate summary files to include. I also looked at the GitHub repo, which looked to have all the described components.

The authors could consider putting a frozen version of the repo on a repository like figshare, which is potentially superior in terms of reproducibility because it can't be deleted by the author.

Analytical approach:

The statistical analyses were valid and robust. When mapping across all the individual yeast genomes, they used a linear mixed effect model (LMM) to account for unequal relatedness of diploid genomes, essentially representing population structure across the families. They even confirmed the issue of structure by showing the extreme inflation for p-values when population structure is ignored when mapping across families. They also mapped QTL at the level of families, at which point the diploid genomes are equally related (in expectation) and share one parent, at which point the LMM is theoretically unnecessary. Furthermore, the approaches for measuring heritability and partitioning genetic effects into additive, dominance, and epistasis were appropriate.

Thank you for the positive feedback above. Regarding Reviewer 2's comment on data availability, we have made the following update described in the Supplement:

Raw barcode sequencing data are available from the NIH Sequence Read Archive as accession PRJNA781980. Additional data (Supplementary data SD1-11) are available in Mendeley data (<https://data.mendeley.com/datasets/96ghpztzvf>). All code used to analyze data, perform statistical analyses, and generate figures is available at Github (<https://github.com/tmatsui2/Matsui-et-al.-2021-Supplemental-Information>).

Suggested improvements:

I don't think any major changes or improvements are necessary for this manuscript. I do have some broad comments, which could be used for minor adjustments to sections of the manuscript.

- Some readers may appreciate some comparison of genetic loci and their effects across environments, which could identify some interesting biology (eg, does the regulating genetic variation make biological sense in terms of potential causal genes and environments?). However, this doesn't really match the main questions tackled in this manuscript and could thus distract from the message on non-additive genetic effects.

Thank you for this comment. While we agree that this is definitely interesting and worthy of future study, we ultimately decided not to include any analysis comparing loci across environments. A concern we have is that because our population of diploids were originally constructed from a panel of only 1,000 F2 haploid segregants, the number of recombination breakpoints are limited. As such, our mapping resolution is often coarse, with loci typically encompassing several genes. Also, given our findings that loci genome-wide can influence the fitness of a strain, we could not confidently say that a locus detected across multiple environments in similar regions along the genome are actually caused by the same variant or even the same gene.

- LMM vs fixed effects regression:

o The authors note that the across-family analysis through LMM didn't yield more loci as might be expected given such a large increase in sample size compared to family-specific scans. This result doesn't actually surprise me that much given the complexity of the overall population structure. The genetic relationship matrix will encompass a coarser population structure, ie, the family blocks of genomes that share an entire parental haplotype, as well as finer-scale differences like the chance differences in shared descent within a family. It's actually very impressive that the LMM does such a good job of reducing the potential for false positives due to this population structure (Fig S7).

o Furthermore, given how effects can vary across genetic backgrounds, which the authors demonstrate nicely, it's not surprising that fewer loci are detected across the whole population. The family-specific scans can be viewed as testing a family-QTL interaction term, so if genetic effects vary across families, it makes sense that a lot would be missed by expecting a consistent QTL effect. There are some QTL detected across families with LMM that don't show up in the family-specific scans (Fig 2A-B). I would speculate that those are potentially weaker QTL that also happen to be fairly consistent across families.

o There is some support for using LMMs even when individuals are expected to be equally related (genomes are essentially randomized with respect to each other), in order to account for the observed imbalance in relationships (PMID: 24906874). I think the impact on power is fairly minimal in this situation, and fixed effects regression is justifiable over LMMs at times, such as due to the computational burden when scanning many families or the overall population is very large.

This is very interesting. To our knowledge, no linkage mapping studies in yeast have accounted for imbalance in relationships when individuals are expected to be equally related, so we believe it is unnecessary.

- The large-scale adjustment for additive effects was very interesting, and not something I was familiar with based on my work largely in inbred mice. Showing the examples (Figs 2F, 3D) very nicely demonstrates its effectiveness for identifying non-additivity.

Thank you for this positive feedback.

- Multiple testing correction:

o The authors describe controlling for the FWER through permutations within a scan-type. Technically there is additional multiple testing occurring across the different analysis types. I think their approach is acceptable though given the goal of looking at non-additive genetic effects. A complex multi-step correction for multiple testing would just distract from this work.

We agree this is an important issue and also appreciate Reviewer 2's recognition that this could be a distraction. Additionally, we point towards our response to a similar comment from Reviewer 1, which contains details about why such an additional layer of multiple testing correction could be difficult to implement. To help address this issue, we included a new analysis identifying loci detected in more families than expected by chance.

- Chromosome IV is a distinct outlier in terms of founder allele balance (Fig S2). Is there anything known about that region in recombinant yeast populations? Maybe it's specific to these parental strains. It does distract from the core message, so I think it's reasonable to not dwell on it.

We have used the same parent strains, BY and 3S, in many previous genetic mapping papers (Taylor et al. 2014. PLOS Genetics; Taylor and Ehrenreich. 2015. PLOS Genetics; Lee et al. 2016. PLOS Genetics; Taylor et al. 2016. Nature Communications; Mullis et al. 2018. Nature Communications; Lee et al. 2019. Genetics). We did not see any biased allele balance on Chromosome IV in any of these papers. We do note that a major effect was seen on Chromosome IV in Mullis et al. 2018. Nature Communications, which was the largest effect locus in that study. However, in a paper that is in revision at Genetics (Schell et al. bioRxiv), we show this locus is due to a de novo mutation in MRP20, an essential subunit of the mitochondrial ribosome. The MRP20 mutation is not present in the segregants used in the current paper and cannot be the cause of the observed imbalance on Chromosome IV.

In the current study, clearly a mitotic recombination must have occurred in some BY/3S diploid cells prior to the generation of BYx3S haploids. We are not sure why this rose to high frequency and it may represent a locus with a biological effect. However, we cannot say for sure and the allelic imbalance does not have a material impact on our story. Thus, we agree that dwelling on this imbalance would detract from the core story of our paper.

Clarity and context:

The manuscript is well-written and I believe easy to follow for a reader with a background in genetics. The Methods section is also nicely written, and covers all the different analyses.

- I wonder if it would be good to add information about the phenotype to the manuscript title? Maybe adjust it to be "The interplay of additivity, dominance, and epistasis on fitness in a diploid yeast cross". I think the findings can be interpreted broadly across phenotypes, so I'm not sure it needs to be adjusted.

We agree and have changed the title as the reviewer suggests.

- The authors mention consolidating loci by using the 95% confidence interval (CI) on QTL position. They don't mention what this interval is based on. CIs for QTL position are usually approximate, using a support interval or parametric bootstrap. I'm sure whatever they used is fine, but it could be good to report it, even if it were the default in Fast-LMM or something like that.

We have added the following sentence to the 3rd paragraph of section "Genome-wide scan for one-locus effect loci" in the methods explaining how the confidence intervals were determined:

"The confidence interval of a locus was defined as the region surrounding the QTL position in which the significance of detection was within three $\times -\log_{10}(\text{p-value})$ of the QTL position (hereafter referred to as "3 LOD drop")."

We also noticed that this is a typo. Loci were consolidated using this 3 LOD drop, not the 95% confidence interval.

- Fig 2E could use more description. The authors show that loci effects detected across families are clearly larger on average than those detected only at the family-level. I think that is very clear, and makes sense, but it's not clear to me what the factor levels are on x-axis. Maybe they've been sorted into bins to show the full spectrum of effects? I think just a little more detail in the captions would suffice.

As part of addressing Reviewer 1's concern about controlling for multiple testing in the family-level genetic mapping, we decided to remove this figure. When we initially submitted this manuscript, we included this figure to demonstrate how family-level scans may enhance statistical power to detect loci with smaller effects or only exhibit effects in few genetic backgrounds. However, the story is more complicated as FaST-LMM and family-level scans likely differ in their abilities to detect true positives for many reasons. Our genetic mapping is quite complex with varying degrees of relatedness among strains within and between families and a potentially large number of segregating loci that vary in their effect sizes, dominance, epistasis, and linkage to each other. There are many reasons why one scan type might detect loci that were not identified in another. To present a more nuanced perspective on these results, we made a new plot (now Fig. 2D) showing the significance of detections across families for all loci that were identified by FaST-LMM or dominance scans or were enriched across family-level scans. We have attempted to make the language regarding this panel clearer and also modified the associated part of the main text section 'Genetic mapping within interrelated families'.

- Minor comments

- o I would slightly rephrase this sentence in the 2nd paragraph of the first section. "Most empirical studies of non-additive genetic effects have focused on haploid or inbred individuals (14–16), which provide higher statistical power to detect loci due to their nominal levels of heterozygosity." "Nominal" has dual meanings, in that it could be referring to a stated level of a summary (ie, nominal p-value vs empirical p-value), whereas I assume the authors mean it as very low levels of heterozygosity. I would consider rephrasing with "minimal" or even "lack of heterozygosity", since residual heterozygosity in inbreds is usually low.

We have changed "nominal" to "minimal", as the reviewer suggests.

- o The words "strain", "genotype", and "diploid" are used as synonyms throughout the manuscript, which is correct given the context, meaning a unique diploid genome. But possibly some readers will be confused or read "genotype" and think specifically of a locus. Just something to consider.

We agree. All places in the manuscript in which strains were referred to as "diploid" or "genotype" were unified as "strains".

o I would consider expanding the title of the Methods section “Correcting for family structure” to something like “Removing additive effects and correcting for family structure”.

The section title has been changed, as the reviewer suggests.

o Technically the coding of the locus effect in Methods section “Family-level scans for one-locus effects” shouldn’t matter; the categorical coding should be equivalent to a count an allele, because either defines two categories. If both homozygotes are present (as in the cross-family analysis), then the coding should matter and can distinguish an additive term. The authors do correctly describe the variable codings for distinguishing additive and non-additive effects in the more complicated model in Methods section “Estimating the fraction of epistatic effect that involves dominance”.

We modified this section to explain how the coding of the locus effect should not matter if it is categorical or numerical for family-level scans. The following text was added to section “Family-level scans for one-locus effects”:

Because diploids within a family are derived from the same *MATa* haploid, they possess only two possible genotype states: BY/BY and BY/3S or BY/3S and 3S/3S, depending on the allele present in the *MATa* progenitor. Although the *locus* term was encoded as a categorical variable here, the coding of the *locus* term should not matter. The categorical coding should be equivalent to a count of an allele because either defines two categories.

• Typos

o The word “strains” is repeated at the end of the first sentence of the second paragraph.

We addressed this.

o Typo in last sentence of the first paragraph of the Methods section “Counting double barcode reads”: “This can result in erroneous double barcode counts and result errors in fitness estimation.” Could re-write as: “This can result in erroneous double barcode codes, leading to errors in fitness estimation.”

We addressed this.

o Typo in the second paragraph of the Methods section “Counting double barcode reads”: “We then fit a linear model between the count of each PCR chimera and the product of these total count of each single barcode within the pool:”. Could re-write as: “We then fit a linear model between the count of each PCR chimera and the product of the total counts of each single barcode within the pool:”

We addressed this.

References:

The authors could add some additional citations for family/population structure and linear mixed effects models for genetic mapping (first paragraph for “Mapping within interrelated families increases statistical power” section). For example, Kang et al 2008 (EMMA), Kang et al 2009 (EMMAX), and Zhou & Stephens 2012 (GEMMA). This would provide greater background for an area that saw a lot of development roughly 10 years ago. The manuscript also isn’t over-stuffed with citations, so a few more would seem fine.

We added a number of citations in this part of the text.

There were also some missing references in the Methods:

- Missing reference in the second sentence of the Methods section “Whole-genome sequencing of parental haploid segregants”.

This method is described in the following papers, which are now referenced.

- J. S. Bloom, I. M. Ehrenreich, W. T. Loo, T.-L. V. Lite, L. Kruglyak, Finding the sources of missing heritability in a yeast cross. *Nature*. **494**, 234–237 (2013).
- M. B. Taylor, I. M. Ehrenreich, Genetic Interactions Involving Five or More Genes Contribute to a Complex Trait in Yeast. *PLoS Genet.* **10**, e1004324 (2014).
- M. N. Mullis, T. Matsui, R. Schell, R. Foree, I. M. Ehrenreich, The complex underpinnings of genetic background effects. *Nat. Commun.* **9**, 3548 (2018).

- Missing reference in the first sentence of the second paragraph of the Methods section “Barcode sequencing”.

We added the reference L. Zhao, Z. Liu, S. F. Levy, S. Wu, Bartender: a fast and accurate clustering algorithm to count barcode reads. *Bioinformatics*. **34**, 739–747 (2018).

- Missing reference in the second sentence of the third paragraph of the Methods section “Barcode sequencing”.

We added the reference F. Li, M. L. Salit, S. F. Levy, Unbiased Fitness Estimation of Pooled Barcode or Amplicon Sequencing Studies. *Cell Syst.* **7**, 521-525.e4 (2018).

- Missing reference in the Methods section “Estimating the fraction of epistatic effect that involves dominance”.

We added the reference S. B. Yu, J. X. Li, C. G. Xu, Y. F. Tan, Y. J. Gao, X. H. Li, Q. Zhang, M. A. S. Maroof, Importance of epistasis as the genetic basis of heterosis in an elite rice hybrid. *Proc. Natl. Acad. Sci. U. S. A.* **94**, 9226–9231 (1997).

Final comment:

I really enjoyed reviewing this manuscript and believe it represents an impressive contribution to the field of complex traits genetics. I recommend its acceptance and publication.

Thank you for the positive feedback.

REVIEWERS' COMMENTS

Reviewer #2 (Remarks to the Author):

Response to Reviewer 1's comments:

The authors have adequately responded to all of Reviewer 1's comments.

1. Reviewer 1 asked the authors to explain why fitness was less correlated between replicate strains within the same experiment compared to strain means across replicate experiments. They show that it stems from the strain's initial frequency within the pool, which will vary more within a single experiment compared to averages across replicate experiments. They have added citations, additional text, and a supplemental figure that demonstrates it in these data.

2. Reviewer 1 had concerns about the dual mapping approaches, in particular how the family scans resulted in many more QTLs while not accounting for the multiple testing burden across families. Both reviewers did acknowledge that the structure of the overall population is complex due to sets of families sharing a single parent strain which does not align with conventional mapping approaches. The authors now more directly address this challenge in the text and have reduced claims that the within family mapping improves power (as it also likely increases the FPR). Furthermore, they have added an additional analysis to the family scans to get at the multiple testing issue by looking for significant levels of co-mapping QTLs across families. They adjusted Figure 2 to reflect this, which nicely shows the overlap in QTLs based on FaST-LMM, dominance, and family scans (as well as the ones supported across families).

3. Reviewer 1 had asked that the following results/inferences be defined for a reduced higher confidence set of QTLs. In response the authors note that the dominance/epistasis findings were based on a separate mapping approach (after subtracting off additive effects), which does not have the added complexity of across and within family mapping. Because there is only one set of results for dominance/epistasis, the significant findings based on permutations do not need to be further refined into a high confidence set.

The authors adjusted Figure 4 as requested.

Response to Reviewer 2's comments:

The authors have adequately responded to all of my comments.

1. They have added text describing all the data and code to the Supplement.

2. They note that comparing QTLs across environments is challenging due to the relatively coarse LD

structure of the mapping population. Given the focus of the paper, it makes sense to not go down that rabbit hole.

3. I had noted the complex population structure and how that affects the various mapping strategies. In response to Reviewer 1, they now address this more head on in the text and have added an additional analysis in which they look for significant overlap in QTLs from the family scans. These are great revisions to the paper.

4. I had asked about chromosome IV which had unusual imbalance in founder contributions, though I did not think it needed to be revised. Their response shows that they have put some thought into it, but do not want to distract from the message of this paper. I agree with that decision.

5. I suggested slightly adjusting the title, which they agreed with.

6. I asked for more details on QTL positional intervals, which they have added.

7. The specifics of the original Fig 2E had confused me, but that has been adjusted as part of the larger revisions to the overall and family QTL scans. I really like the new Fig 2D, which really highlights how much the signal varies between the mapping methods. Really strong signals tend to be seen across the methods, but there are QTLs only seen within some families and some only seen in the overall scans. This potentially reflects genetic effects that are only seen in certain backgrounds or subtle effects that are only significant when using all the data.

8. They have addressed all minor typos and missing references that I noticed.

Final comment: This is a great paper, and the revision has made it even more accessible. I look forward to pointing others to it once it is published.

January 26, 2022

To the Reviewers,

Thank you for your thoughtful assessment of our manuscript. You had no additional requests, but nonetheless we have included a point-by-point response below. Thank you again.

Sincerely,

Takeshi Matsui, Martin Mullis, Sasha Levy, and Ian Ehrenreich

Reviewer #2 (Remarks to the Author):

Response to Reviewer 1's comments:

The authors have adequately responded to all of Reviewer 1's comments.

1. Reviewer 1 asked the authors to explain why fitness was less correlated between replicate strains within the same experiment compared to strain means across replicate experiments. They show that it stems from the strain's initial frequency within the pool, which will vary more within a single experiment compared to averages across replicate experiments. They have added citations, additional text, and a supplemental figure that demonstrates it in these data.

Thank you for being satisfied with our response.

2. Reviewer 1 had concerns about the dual mapping approaches, in particular how the family scans resulted in many more QTLs while not accounting for the multiple testing burden across families. Both reviewers did acknowledge that the structure of the overall population is complex due to sets of families sharing a single parent strain which does not align with conventional mapping approaches. The authors now more directly address this challenge in the text and have reduced claims that the within family mapping improves power (as it also likely increases the FPR). Furthermore, they have added an additional analysis to the family scans to get at the multiple testing issue by looking for significant levels of co-mapping QTLs across families. They adjusted Figure 2 to reflect this, which nicely shows the overlap in QTLs based on FaST-LMM, dominance, and family scans (as well as the ones supported across families).

Thank you for raising this issue and for being satisfied with our response.

3. Reviewer 1 had asked that the following results/inferences be defined for a reduced higher confidence set of QTLs. In response the authors note that the dominance/epistasis findings were based on a separate mapping approach (after subtracting off additive effects), which does not have the added complexity of across and within family mapping. Because there is only one set of results for dominance/epistasis, the significant findings based on permutations do not need to be further refined into a high confidence set.

Thank you for being satisfied with our response.

The authors adjusted Figure 4 as requested.

Thank you for being satisfied with our response.

Response to Reviewer 2's comments:

The authors have adequately responded to all of my comments.

1. They have added text describing all the data and code to the Supplement.

Thank you for being satisfied with our response.

2. They note that comparing QTLs across environments is challenging due to the relatively coarse LD structure of the mapping population. Given the focus of the paper, it makes sense to not go down that rabbit hole.

Thank you for being satisfied with our response.

3. I had noted the complex population structure and how that affects the various mapping strategies. In response to Reviewer 1, they now address this more head on in the text and have added an additional analysis in which they look for significant overlap in QTLs from the family scans. These are great revisions to the paper.

Thank you for your input and for being satisfied with our response.

4. I had asked about chromosome IV which had unusual imbalance in founder contributions, though I did not think it needed to be revised. Their response shows that they have put some thought into it, but do not want to distract from the message of this paper. I agree with that decision.

Thank you for being satisfied with our response.

5. I suggested slightly adjusting the title, which they agreed with.

Thank you for the suggestion.

6. I asked for more details on QTL positional intervals, which they have added.

Thank you for being satisfied with our response.

7. The specifics of the original Fig 2E had confused me, but that has been adjusted as part of the larger revisions to the overall and family QTL scans. I really like the new Fig 2D, which really highlights how much the signal varies between the mapping methods. Really strong signals tend to be seen across the methods, but there are QTLs only seen within some families and some only seen in the overall scans. This potentially reflects genetic effects that are only seen in certain backgrounds or subtle effects that are only significant when using all the data.

Thank you for raising this issue in our original submission and for being satisfied with our response.

8. They have addressed all minor typos and missing references that I noticed.

Thank you for the constructive input.

Final comment: This is a great paper, and the revision has made it even more accessible. I look forward to pointing others to it once it is published.

Thank you! We really appreciate your insights. They definitely improved our paper.